# A locally solvent-tethered polymer electrolyte for long-life lithium metal batteries

Yanfei Zhu [1,3], Zhoujie Lao[1,3], Mengtian Zhang[1,3], Tingzheng Hou [1] ✉, Xiao Xiao[1], Zhihong Piao[1], Gongxun Lu [1], Zhiyuan Han[1], Runhua Gao[1], Lu Nie[1], Xinru Wu[1], Yanze Song[1], Chaoyuan Ji[1], Jian Wang [2] & Guangmin Zhou [1] ✉

Solid polymer electrolytes exhibit enhanced Li$^+$ conductivity when plasticized with highly dielectric solvents such as N,N-dimethylformamide (DMF). However, the application of DMF-containing electrolytes in solid-state batteries is hindered by poor cycle life caused by continuous DMF degradation at the anode surface and the resulting unstable solid-electrolyte interphase. Here we report a composite polymer electrolyte with a rationally designed Hofmann-DMF coordination complex to address this issue. DMF is engineered on Hofmann frameworks as tethered ligands to construct a locally DMF-rich interface which promotes Li$^+$ conduction through a ligand-assisted transport mechanism. A high ionic conductivity of $6.5 \times 10^{-4}$ S cm$^{-1}$ is achieved at room temperature. We demonstrate that the composite electrolyte effectively reduces the free shuttling and subsequent decomposition of DMF. The locally solvent-tethered electrolyte cycles stably for over 6000 h at 0.1 mA cm$^{-2}$ in Li||Li symmetric cell. When paired with sulfurized polyacrylonitrile cathodes, the full cell exhibits a prolonged cycle life of 1000 cycles at 1 C. This work will facilitate the development of practical polymer-based electrolytes with high ionic conductivity and long cycle life.

Solid polymer electrolytes (SPEs) are recognized as promising candidates for achieving next-generation solid-state batteries due to their accessible processability and high safety[1]. However, the sluggish segmental motion and limited solvating ability of the polymer matrix lead to unsatisfactory ionic conductivity of SPEs at room temperature ($10^{-8}$ to $10^{-5}$ S cm$^{-1}$)[2,3]. High dielectric solvents including N-methyl-2-pyrrolidone (NMP)[4], dimethyl sulfoxide (DMSO)[5], and N,N-dimethylformamide (DMF)[6] have been utilized in SPEs as plasticizers to facilitate improved transport kinetics. Among the investigated solvents, DMF with high relative permittivity ($\varepsilon_r = 36.7$) and low viscosity (0.82 mPa s) at room temperature has proven to be highly effective for boosting ionic conductivity[7,8]. The carbonyl oxygen (C=O) of DMF directly coordinates with Li$^+$, which helps dissociate the ion pair of lithium salts. This DMF regulated Li$^+$ coordination environment reduces the correlated motion of Li$^+$ and the anion, and promotes net Li$^+$ fluxes. Unfortunately, it is found that DMF can migrate together with Li$^+$ towards the Li metal anode, leading to serious side reactions at the electrode–electrolyte interface[9]. The uncontrolled decomposition and continuous depletion of DMF are highly detrimental to the stability of the Li metal anode as well as the durability of SPEs.

Additive engineering has been a cost-effective and practical approach to mitigate the adverse effects of DMF in SPEs[10]. Functional polymer (e.g., poly(acrylic acid)) has been explored as an organic additive to induce a robust solid-electrolyte interphase (SEI), and suppress the interfacial side reactions between DMF and Li metal. However, despite the improved interfacial stability and cycle life, DMF

[1]Tsinghua-Berkeley Shenzhen Institute & Tsinghua Shenzhen International Graduate School, Shenzhen 518055, PR China. [2]Canadian Light Source, Saskatoon S7N 2V3, Canada. [3]These authors contributed equally: Yanfei Zhu, Zhoujie Lao, Mengtian Zhang. ✉e-mail: tingzhenghou@sz.tsinghua.edu.cn; guangminzhou@sz.tsinghua.edu.cn

is virtually diluted and cannot sufficiently dissociate the Li salts when excessive organic additive filled, resulting in compromised ionic conductivity[11]. Composite polymer electrolytes (CPEs) with inorganic additive have been investigated with improved ionic conductivity by reducing the polymer crystallinity and providing multiple/or synergistic Li+ conductive pathways[12]. Recently, dielectric ceramic materials such as $Li_{6.75}La_3Zr_{1.75}Ta_{0.25}O_{12}$ (ref. 13), $Li_{1.4}Al_{0.4}Ti_{1.6}(PO_4)_3$ (ref. 14), and $BaTiO_3$–$Li_{0.33}La_{0.57}TiO_{3-x}$ (ref. 15) and have been reported to further promote salt dissociation while serving as active sorbents for DMF. However, previous approaches to additive engineering have mainly focused on addressing either ionic conductivity or electrochemical stability. The synergetic effect of DMF modulation and Li+ transport mechanism among the complex components remains unclear, which substantially impedes the design of high-performance and durable polymer electrolytes. In brief, an effective strategy to simultaneously improve both aspects needs to be developed for the ultimate application of SPEs.

Based on our previous efforts in the design of nano-additive for CPEs[16], we propose to design a DMF-containing coordination complex as a dual-functional additive. By engineering DMF into a suitable coordination complex, surface-tethered DMF ligands are expected to facilitate ion transport by a locally solvent-rich environment while minimizing undesirable DMF migration. Among reported coordination complexes, layered Hofmann-framework materials are known for their metal site-rich feature and tunable geometry by incorporating appropriate ligands (e.g., $NH_3$, $H_2O$, and organic molecules) on the metal sites[17–19]. A wide range of functionalized Hofmann materials with unique physicochemical properties can be developed through fine-tuning ligands and ligand exchange. This feature provides the opportunity to locally confine and retain functional solvents, including DMF, within the electrolyte phase[20]. Moreover, open metal sites of the Hofmann framework are able to dissociate Li salt through the metal–anion interaction[21,22]. Therefore, integrating the Hofmann framework with DMF as ligands is a promising approach to construct a dual-functional additive.

Here we demonstrate the simultaneous improvement in ionic conductivity and electrochemical stability for CPEs by the incorporation of DMF ligand-exchanged Hofmann frameworks. A locally-confined DMF-rich environment at additive–polymer interfaces of CPEs is constructed by a Hofmann-DMF $(Ni(DMF)_2Ni[CN]_4$, denoted as Ni-DMF). The confinement and reduced consumption of DMF leads to a stable inorganic-rich SEI with a significant decrease in DMF induced products and a durable CPE. As a result, the designed CPE cycled stably against Li metal electrodes for over 6000 h with an overvoltage of 64 mV. Meanwhile, the constructed solvent-tethered Ni-DMF–electrolyte interfaces serve as rapid Li+ transport pathways and promote Li+ conduction kinetics through a ligand-assisted transport mechanism, leading to a high room temperature ionic conductivity $(6.5 \times 10^{-4}\,S\,cm^{-1})$. Furthermore, Ni-DMF helps immobilize anions and exhibits a high Li+ transference number $(t_{Li^+})$ of 0.71. These features enable Li||sulfurized polyacrylonitrile (Li||SPAN) cells to operate for 1000 cycles at 1 C with a capacity decay of 0.04% per cycle and a Li||SPAN pouch cell with an areal capacity of 1.9 mAh cm$^{-2}$ (47 mAh with sulfur loading of 50 mg) to operate for 35 cycles, all of which stand among the state-of-the-art performance in solid-state lithium–sulfur batteries (ssLSBs).

## Results

### Integrated Ni-DMF complex design

A precursor Hofmann-$H_2O$ complex $(Ni(H_2O)_2Ni[CN]_4·xH_2O$, denoted as Ni-$H_2O$) was synthesized by assembling planar anionic $[Ni(CN)_4]^{2-}$ with $Ni^{2+}$ ions, which link neighboring cyanogen groups at their N ends (Supplementary Fig. 1)[18]. The $Ni^{2+}$ sites bonding with two coordinated $H_2O$ molecules further complete the octahedral coordination geometry. The resulting crystal structure is an extended two-dimensional

(2D) network stacking along the [001] axis, with absorbed $H_2O$ molecules in the interlayer region. The absorbed $H_2O$ is unstable and can be removed by external management, resulting in abundant interstitial space[23]. With this in mind, a thermal treatment process for Ni-$H_2O$ was carried out to obtain the intermediate products $(Ni(H_2O)_2Ni[CN]_4$, denoted as Ni-activated) with open channels created for efficient ligand exchange. Scanning electron microscopy (SEM) and transmission electron microscopy (TEM) images show that Ni-$H_2O$ and Ni-activated exhibit similar nano-sheet morphology with a diameter of approximately 200 nm (Supplementary Fig. 2). $N_2$ adsorption/desorption isotherms demonstrate the removal of absorbed $H_2O$, as evidenced by an uplift at low P/$P_0$ for Ni-activated, which corresponds to an augmentation of interstitial space (Supplementary Fig. 3). This conversion is further supported by Brunauer–Emmett–Teller (BET) surface area $(S_{BET})$ calculation that the Ni-activated achieves a higher $S_{BET}$ of 48.4 m$^2$ g$^{-1}$ than Ni-$H_2O$ (27.6 m$^2$ g$^{-1}$). With an enlarged open channel, the coordinated $H_2O$ was subsequently substituted with DMF through the ligand exchange process with excessive DMF filling up (Fig. 1a)[2]. The obtained Ni-DMF structure consists of 2D corrugated layers stacking along the [200], achieving both enlarged lattice spacing (7.63 Å) and $S_{BET}$ of 59.7 m$^2$ g$^{-1}$. Powder X-ray diffraction (PXRD) results for Ni-DMF show a (200) crystal peak at around 12°, which coincides well with the simulated $Ni(DMF)_2Ni[CN]_4$ diffraction pattern (Supplementary Fig. 4)[24].

Scanning transmission X-ray microscope (STXM) energy-stack imaging technique[25] and extracted X-ray absorption near edge structure (XANES) spectra of O K-edge further reveal the ligand exchange process (Fig. 1b). Stack data were analyzed using the principle component analysis and cluster analysis to obtain the spatial distribution of O containing species, presented as color-coded mapping images based on changes in optical densities (Supplementary Figs. 5–7). The O K-edge spectra extracted from each sample stack region are shown in Fig. 1c. Compared with the Ni-$H_2O$ sample, the inorganic O (H–O–H) signals of Ni-activated located at 536 and 538 eV are weakened due to the elimination of absorbed $H_2O$ molecules[26]. In the case of Ni-DMF, the emergence of a strong organic O (C=O in DMF) signal (532 eV) instead of H–O–H signals indicates efficient O substitution[27]. These results are consistent with attenuated total reflectance-Fourier transform infrared spectra (ATR-FTIR), which show that H–O–H signals are replaced with C=O signals. (Supplementary Fig. 8).

We further conducted a comprehensive analysis of the local structure of Ni-DMF using both theoretical calculations and extended X-ray absorption fine structure (EXAFS) analyses. Density functional theory (DFT) calculation results demonstrate that the formation of the Ni-DMF structure is overall exergonic, as evidenced by a more negative adsorption Gibbs free energy (ΔGr) under the excessive DMF solvent environment (Fig. 1d). The k$^3$-weighted Fourier transform of EXAFS reveals the local structure change of Ni site (Supplementary Fig. 9). The first coordinated shells are assigned to the Ni–C or Ni–N scattering paths[16]. The relatively stronger intensity of Ni–O oscillation in Ni-DMF reveals its more ordered local structure around the Ni site[28]. Notably, Ni-DMF exhibits a larger Ni–O bond length (1.77 Å) than that of Ni-$H_2O$ (1.74 Å), revealing the weaker interaction between Ni and DMF ligand.

To further investigate the interaction between the Ni site and its surrounding ligands, we conducted STXM XANES analyses at the Ni L-edge (Fig. 1e). Both samples show three absorption peaks in the $L_3$-edge region where the peak located at ~859 eV is assigned to metal–ligand interaction of the Ni coordination[29]. Compared with Ni-$H_2O$, Ni-DMF witnesses a decrease in peak intensity, indicating a less charge exchange interaction between Ni and DMF[30]. The highest occupied molecular orbital analysis (HOMO, insets in Fig. 1e) further illustrates the local electronic structural difference between Ni-$H_2O$ and Ni-DMF. Ni-$H_2O$ shows a more shared electron distribution, whereas Ni-DMF mostly exhibits electrostatic interactions with comparatively less charge transfer.

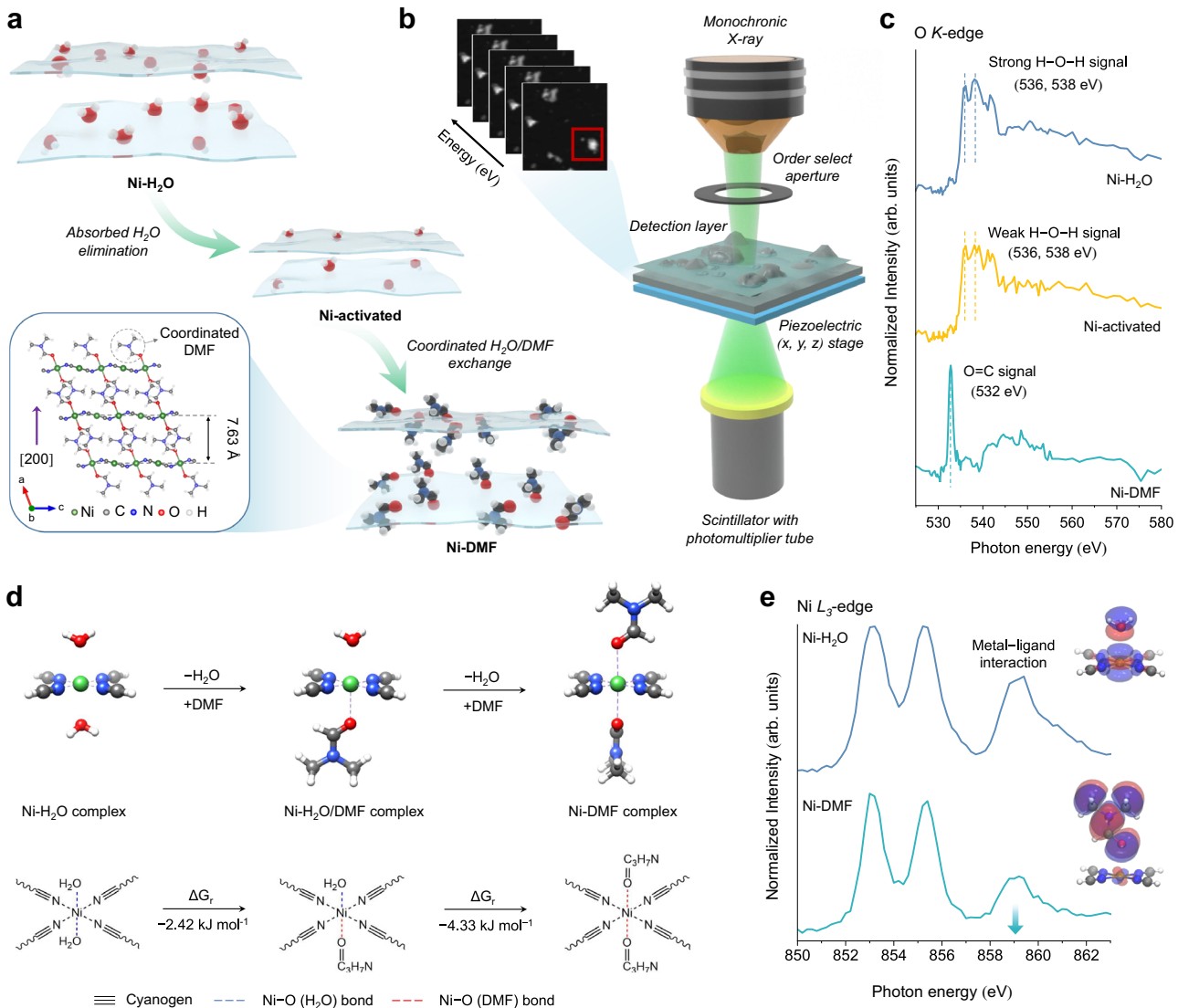

**Fig. 1 | Experimental and theoretical study on the ligand-exchanged processes and the structure-property correlations. a** Schematic illustration of the ligand-exchanged processes and the crystal structure of Ni-DMF. **b** Stack analyses of Ni-DMF detected by STXM. **c** Aligned average values of O K-edge spectra of Ni-H₂O, Ni-activated, and Ni-DMF. **d** Coordination structures and adsorption Gibbs free energy (ΔGr) change during ligand-exchanged processes calculated using DFT. **e** XANES analyses of Ni L₃-edge for Ni-H₂O and Ni-DMF and corresponding charge interaction between Ni and surrounding ligands.

## Li⁺ transport mechanism in CPE

To investigate the ionic conduction behavior of Ni-DMF, the materials were incorporated into a LiFSI/poly(vinylidene fluoride-co-hexafluoropropylene) (PVDF-HFP) electrolyte (donated as LPE) to yield a CPE (donated as LPE@Ni-DMF). LPE@Ni-DMF exhibits a much higher ionic conductivity ($6.5 \times 10^{-4}$ S cm⁻¹) at room temperature than that of LPE ($2.4 \times 10^{-4}$ S cm⁻¹) and other reported PVDF-based electrolytes (Supplementary Fig. 10 and Table 1). Furthermore, the $t_{Li^+}$ of LPE@Ni-DMF increases from 0.44 (LPE) to as high as 0.71 (Supplementary Fig. 11). To understand the contribution of Ni-DMF to the improved ionic conduction performance, molecular dynamics (MD) simulations were performed to reveal the Li⁺ transport mechanism on the atomistic scale (Supplementary Fig. 12). Extracted local snapshots show the structural evolution of key species in LPE@Ni-DMF (Fig. 2a). At the initial stage (0 ns), the DMF ligand binds to the Ni site on the surface via Ni−O(DMF) coordination. Subsequently, the DMF detethers from the Ni-DMF surface, and enters the first solvation shell of a nearby Li⁺ to form a Li−O(DMF) coordinated state at 0.76 ns. Importantly, the recoordination of Ni−O(DMF) is captured at 36.79 ns where the DMF

ligand decoordinates with Li⁺ and re-tethers to the original Ni site. The observation of the reverse process indicates that the surface-tethered DMF can facilitate ion motion while barely diffusing with Li⁺. By extending the simulation time, we observed rotational behaviors of the Li⁺-coordinating DMF ligand during the Li⁺ transport process (Fig. 2b and Supplementary Fig. 13). Therefore, a locally-confined DMF-rich interface between the Ni-DMF and its surrounding electrolyte phase is theoretically verified[31,32]. We also note that FSI⁻ can be immobilized onto the framework (Supplementary Fig. 14). The motion of the paring Li⁺ can thus be uncorrelated from the anion, contributing to an improved Li⁺ conductivity and transference number[33]. As a comparison, in LPE, it is observed that the coordinating DMF tends to diffuse together with Li⁺ for a longer distance (Fig. 2c). We further quantitatively distinguished the ionic conduction mechanism of LPE@Ni-DMF and LPE by calculating the characteristic residence time (τ) for coordinating species moving with Li⁺ (Supplementary Fig. 15). We observe a notable reduction of 60% in the residence time for both Li⁺/DMF and Li⁺/FSI⁻ pairs within LPE@Ni-DMF compared to pristine LPE, in accordance with the lower activation energy (Eₐ) of LPE@Ni-DMF (0.147 eV)

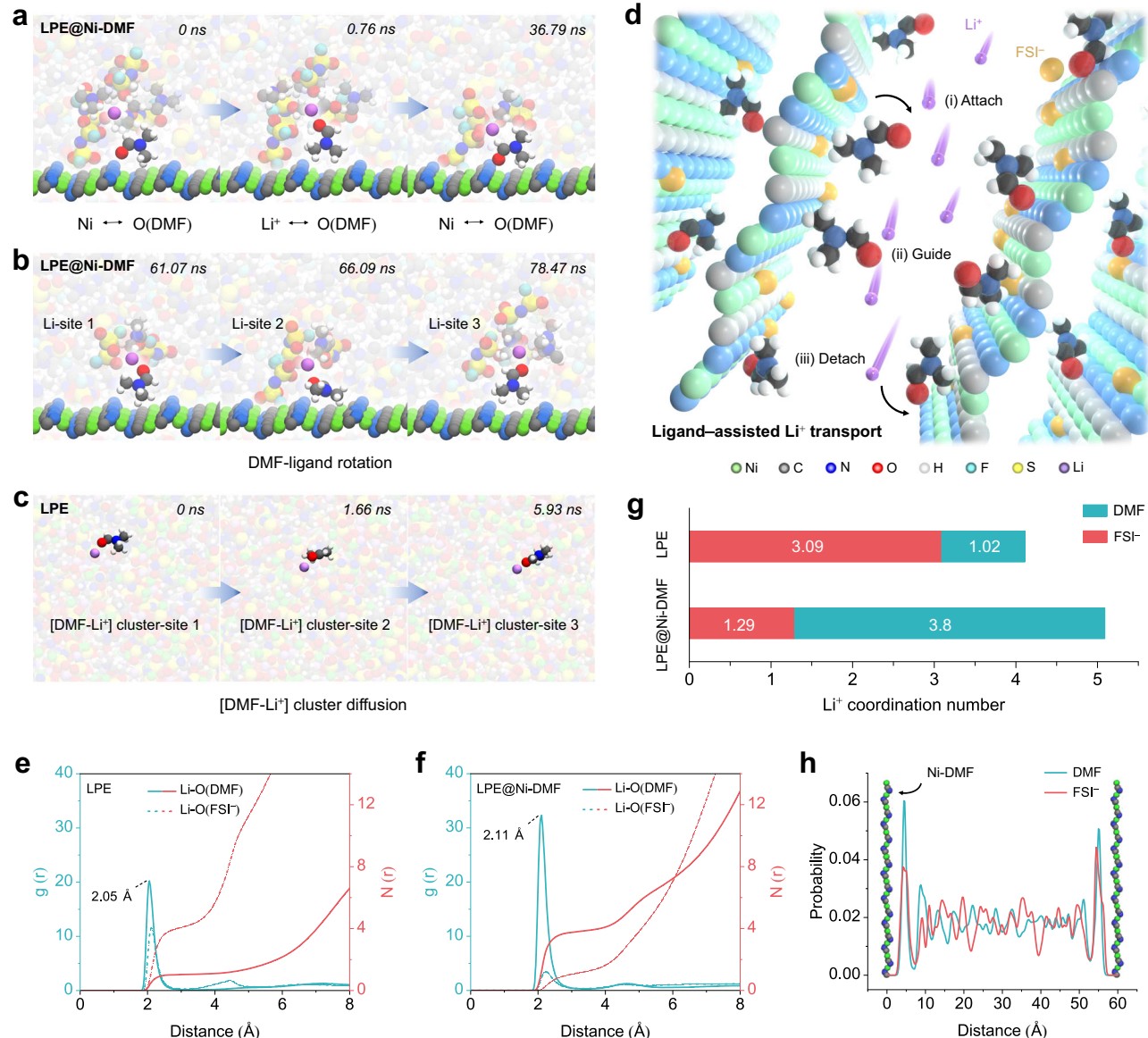

**Fig. 2 | Li⁺ transport mechanism in LPE@Ni-DMF.** Snapshots of MD simulation of the interaction between Li⁺ and DMF ligand (**a**) and local motion trail of the DMF ligand during Li⁺ transfer (**b**). **c** Snapshots of the MD simulation of [DMF-Li⁺] cluster diffusion in LPE. **d** Schematic illustration of the ligand-assisted Li⁺ transport mechanism where Li⁺ is attached, guided, and then detached with DMF ligands at

the locally-confined DMF-rich interfaces. RDFs of Li⁺/DMF and Li⁺/FSI⁻ pairs in LPE (**e**) and LPE@Ni-DMF (**f**). **g** Integrated coordination number of Li⁺ with DMF and FSI⁻ in LPE and LPE@Ni-DMF, respectively. **h** Probability distribution of DMF and FSI⁻ between two separate Ni-DMF nanosheets in LPE@Ni-DMF.

than that of LPE (0.218 eV; Supplementary Fig. 16). The results demonstrate a less correlated cation–anion motion (higher $t_{Li^+}$), and a less sluggish ion transport mechanism with more Li⁺ hopping (higher conductivity) in LPE@Ni-DMF[34]. This trend is in well agreement with our experimental measurements. The diffusion length of DMF co-migrating with Li⁺ can be determined utilizing the equation:

$$L = \sqrt{6 D_{DMF} \tau} \qquad (1)$$

Since the diffusion coefficients for DMF ($D_{DMF}$) and FSI⁻ ($D_{FSI^-}$) remain relatively constant in LPE and LPE@Ni-DMF (Supplementary Fig. 17), both the residence time and diffusion length of DMF moving concurrently with Li⁺ exhibit significant decrease within LPE@Ni-DMF, which mitigates the DMF shuttling. Therefore, Li⁺ conduction in

LPE@Ni-DMF follows a unique ligand-assisted Li⁺ transport mechanism at the Ni-DMF−electrolyte interfaces (Fig. 2d).

To further illustrate how the locally DMF-rich environment leads to a favorable Li⁺ transport mechanism, radial distribution functions (RDFs) of the Li⁺ coordination shell were calculated for bulk LPE and the interface in LPE@Ni-DMF. The two peaks of LPE, located at approximately 2 Å are assigned to Li−O(DMF) and Li−O(FSI⁻) pairs, corresponding to an integrated coordination number (CN) of 1.02 and 3.09, respectively (Fig. 2e, g)[35]. In contrast, the dominant peak of the Li−O(DMF) pair in LPE@Ni-DMF suggests that more DMF molecules participate in Li⁺ solvation, resulting in a larger CN of 3.8 and a larger Li−O(DMF) distance of 2.11 Å in LPE@Ni-DMF compared to that in LPE (2.05 Å) (Fig. 2f, g). The DMF ligands of Ni-DMF alter the Li⁺ coordination environment in LPE, leading to an increase in the number of solvent molecules surrounding Li⁺ and consequently a longer Li−O(DMF) bond length. This reduces the interaction between Li⁺ and

DMF molecules, including those trapped in the PVDF-HFP matrix. Additionally, the reduced peak of Li−O(FSI⁻) pair indicates more dissociation between Li⁺ and FSI⁻, which arises from the enhanced solvation effects of DMF ligands and the immobilization of FSI⁻ of Ni sites (Supplementary Fig. 18). Overall, both DMF and FSI⁻ tend to concentrate at the interface, inducing a DMF-rich and FSI⁻-adsorbed local environment (Fig. 2h).

## Properties of LPE@Ni-DMF

LPE@Ni-DMF exhibits a comparable thickness of ~130 µm, yet a more compact structure than LPE, as evidenced by SEM images (Supplementary Fig. 19). To further investigate the internal architecture and component distribution of LPE@Ni-DMF, X-ray computed tomography (XCT) was employed (Fig. 3a). The Ni-DMF nanosheets (green

region in the split rendering map) are uniformly dispersed in PVDF-HFP and LiFSI matrix, thereby creating continuous and efficient Li⁺ transfer interfaces between Ni-DMF and electrolyte phases (Supplementary Movie 1). The three-dimensional (3D) reconstruction results reveal that the Ni-DMF compactly integrate with the LPE matrix, leaving behind interconnected micron-level particle (Fig. 3b and Supplementary Movie 2). This unique spatial configuration is further supported by magnified SEM images (Fig. 3c and Supplementary Fig. 20) and corresponding energy dispersive spectrometer (EDS) mapping (Supplementary Fig. 21). Consequently, LPE@Ni-DMF exhibits a three-fold increase in Young's modulus, reaching 13.3 MPa compared to LPE (Supplementary Fig. 22).

⁷Li solid-state nuclear magnetic resonance (ssNMR) spectra were utilized to investigate the macroscopic environment of Li⁺. In the case

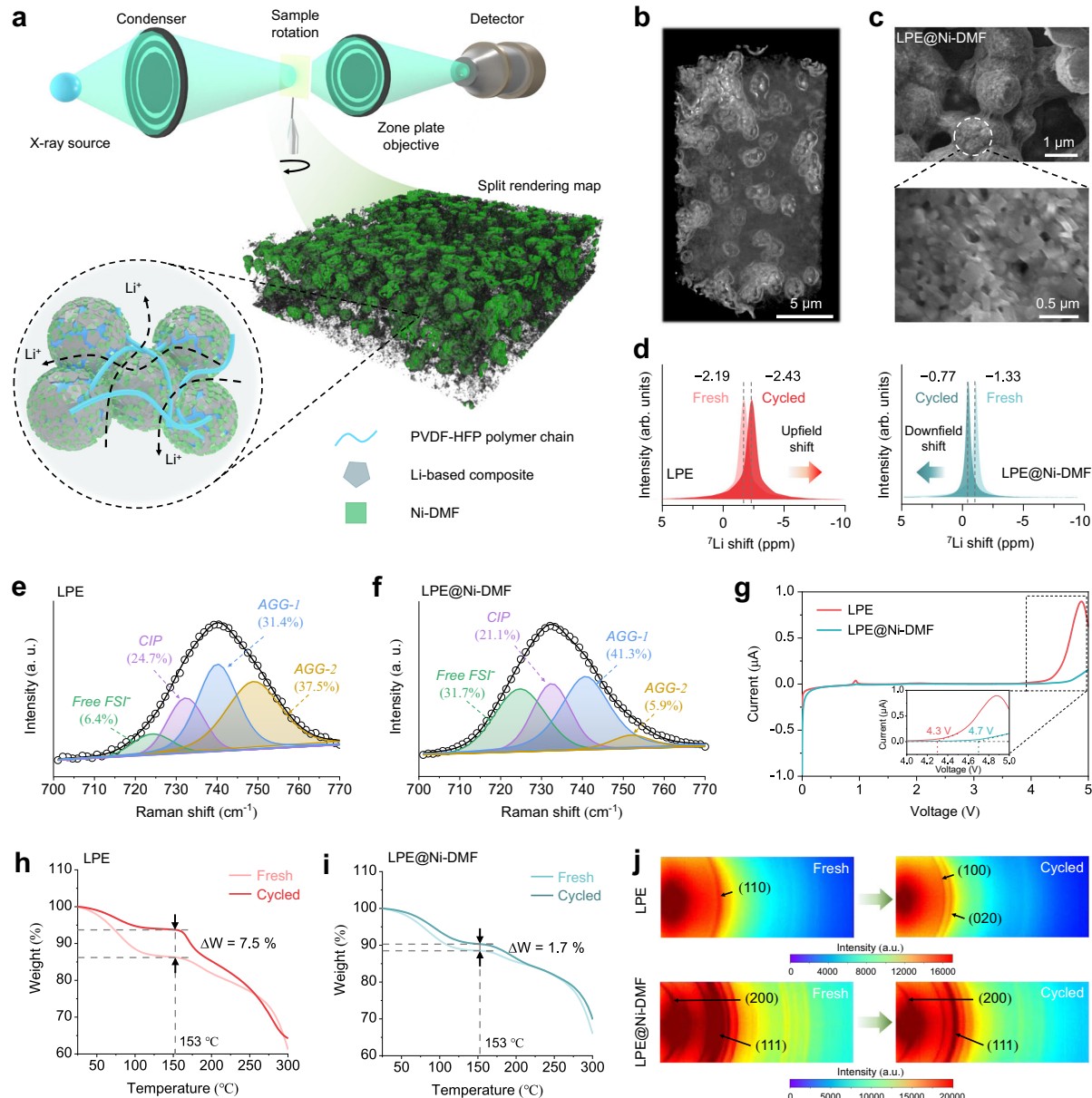

**Fig. 3 | Physicochemical properties and durability analyses of LPE@Ni-DMF.**
**a** Schematic illustration of LPE@Ni-DMF XCT and the internal architecture. The rendered green region represents the uniformly dispersed Ni-DMF nanosheets. **b** 3D bulk reconstruction of LPE@Ni-DMF. **c** SEM images of the internal structure in LPE@Ni-DMF. **d** ⁷Li ssNMR spectra of LPE and LPE@Ni-DMF under fresh and cycled states. Raman spectra of LPE (**e**) and LPE@Ni-DMF (**f**). CIP, contact ion pairs; AGGs,

aggregate clusters. **g** LSV curves of asymmetric Li‖SS cells with a scanning rate of 1 mV s⁻¹. The inset is the magnified image at the selected voltage range from 4 to 5 V. Relative weight change of DMF before and after cycling in LPE (**h**) and LPE@Ni-DMF (**i**) detected by TG analyses. **j** 2D WAXS mapping of LPE and LPE@Ni-DMF under fresh and cycled states.

of LPE where ions are strongly aggregated, a broad resonance is observed due to poor ionic mobility (Fig. 3d)[16,36]. Notably, the narrower $^7$Li full-width of LPE@Ni-DMF than LPE indicates faster Li$^+$ motion facilitated by ligand-assisted transport. Furthermore, a significant difference is observed between cycled samples (galvanostatic plating/stripping tests at 0.1 mA cm$^{-2}$ and 0.1 mAh cm$^{-2}$ per half cycle for 100 h in Li||Li symmetric cells at room temperature), where the $^7$Li signal of the cycled LPE is upfield shift (from −2.19 to −2.43 ppm) compared to a downfield shifted of $^7$Li signal (from −1.33 to −0.77 ppm) of the cycled LPE@Ni-DMF (Supplementary Fig. 23). This trend suggests a reduced deshielding effect on Li$^+$, signifying an enhanced Li salt dissociation[37]. ATR-FTIR tests of LPE and LPE@Ni-DMF membranes were performed to evaluate the trace of C=C double bonds[9]. It is evident that the spectrum of LPE@Ni-DMF shows a more prominent C=C characteristic peak at 1703 cm$^{-1}$ than that of LPE, demonstrating that the Ni-DMF can induce enhanced dehydrofluorination of PVDF-HFP, thereby improving the ionic conduction at the interface of the composite material (Supplementary Fig. 24). Raman analysis was performed to detect the FSI$^-$ anion states[15]. From Fig. 3e, f, the content of free FSI$^-$, contact ion pairs (CIP), and aggregate clusters (AGGs) of LPE are 6.4%, 24.7%, and 68.9%, respectively, while the corresponding values for LPE@Ni-DMF are 31.7%, 21.1%, and 47.2%, respectively, suggesting that the solvation structures can be altered by the addition of Ni-DMF and hence the formation of more mobile Li$^+$.

The electrochemical potential window of electrolytes is an important criterion for evaluating practicality. The high voltage stability of LPE and LPE@Ni-DMF was tested by linear sweep voltammetry (LSV). The LPE exhibits a sharp oxidizing peak beginning at approximately 4.3 V, while the LPE@Ni-DMF begins to oxidize at over 4.7 V, indicating a high antioxidative capability of LPE@Ni-DMF (Fig. 3g). Durability is another crucial criterion to evaluate the practical performance of SPEs[38]. Due to the high reactivity between DMF and Li metal (Supplementary Figs. 25, 26), the shuttling of [DMF-Li$^+$] clusters can lead to a gradual depletion of DMF in LPE. To illustrate the concentration evolution of DMF in a symmetric cell model, the finite element method (FEM) simulation was performed (Supplementary Fig. 27)[39]. Under an applied external electric field, DMF migrates to the anode and reacts with Li metal, resulting in a continuous decrease in DMF concentration in LPE (Supplementary Fig. 28). However, this change is notably less severe in LPE@Ni-DMF due to the stronger DMF anchoring ability of Ni sites compared to PVDF chains (Supplementary Figs. 28, 29). This trend was further experimentally verified by thermal gravimetric (TG) analysis. The relative weight change after cycling (ΔW) reduced from 7.5 wt% for LPE (Fig. 3h) to 1.7 wt% for LPE@Ni-DMF (Fig. 3i).

The diffraction patterns of electrolyte membranes were recorded using 2D wide-angle X-ray scattering (WAXS) to investigate the impact of DMF content evolution on electrolyte properties. For fresh LPE, the crystallographic orientation exhibits a single diffraction peak at 14.9°, corresponding to the (110) crystalline phase of PVDF (Supplementary Fig. 30)[40]. This sparseness indicates an incomplete crystallization of PVDF units in the copolymer due to the presence of DMF, as validated by the highly centered (110) facet in the 2D WAXS concentrated pattern (Fig. 3j). After cycling, two additional crystalline peaks appear at 13.3° (100) and 14.4° (020), respectively, suggesting an increased crystallinity in LPE due to continuous consumption of DMF[9]. The fresh and cycled LPE@Ni-DMF exhibit the same diffraction peaks mainly assigned to Ni-DMF, indicating the robust structural stability of Ni-DMF during the electrochemical process. Additionally, the XRD pattern of cycled LPE@Ni-DMF shows a similar full width at half-maximum of the amorphous area as that of fresh LPE@Ni-DMF, indicating the predominant amorphous structure in LPE@Ni-DMF is well maintained (Supplementary Fig. 31). Consequently, LPE@Ni-DMF with superior durability maintained a higher ionic conductivity of $5.7 \times 10^{-4}$ S cm$^{-1}$ after cycling than that of cycled LPE ($9.5 \times 10^{-5}$ S cm$^{-1}$; Supplementary Fig. 32).

## Interfacial stability of LPE@Ni-DMF with Li metal

The confinement of DMF not only enhances the durability of LPE@Ni-DMF by mitigating DMF shuttling and degradation, but also prevents the irreversible loss of the anode active material. FEM simulations depict the adverse effect of DMF on the Li metal (Supplementary Fig. 33). Benefitting from the confinement effect of DMF, LPE@Ni-DMF produces a relatively flat surface geometry of Li metal. However, Li metal suffers from serious attacks with uneven corrosion depth in the case of LPE, leading to poor interfacial contact and unstable SEI between electrolyte and electrode. As a result, the Li|LPE@Ni-DMF|Li cell stably cycles for 6250 h with a low overvoltage of 64 mV, whereas the cell using LPE displays increasing overvoltage and interfacial resistance after only 500 h of cycling (Fig. 4a and Supplementary Fig. 34). In addition, the Li|LPE@Ni-DMF|Li cell exhibits a critical current density (CCD) of 1.0 mA cm$^{-2}$, which is much higher than that with LPE (0.3 mA cm$^{-2}$; Fig. 4b). It should be noted that the SEM and corresponding EDS mappings show that the Li metal after Li|LPE@Ni-DMF|Li cell cycling has a flat surface morphology without any detectable Ni species, indicating the good electrochemical stability of Ni-DMF on Li metal (Supplementary Fig. 35).

Exchange current density (ECD) tests and X-ray photoelectron spectroscopy (XPS) were conducted to illustrate the different SEI characteristics of LPE and LPE@Ni-DMF. The cell with LPE@Ni-DMF delivers a higher ECD (0.69 mA cm$^{-2}$) than that with LPE (0.24 mA cm$^{-2}$), indicating promoted Li$^+$ transport ability through SEI (Fig. 4c), while a low value of LPE indicates sluggish Li$^+$ conduction kinetics[41]. The SEI composition is found responsible for the difference. The *O1s* XPS spectra obtained through sputtering show that the SEI of LPE (LPE-SEI) displays a clear and high atomic ratio of organic O (−(R−O)$_x$−) signal, while that of LPE@Ni-DMF (LPE@Ni-DMF-SEI) exhibits a decreased signal for the species (Supplementary Fig. 36)[42]. This suggests that the decomposition of shuttled DMF on the Li metal surface is alleviated by LPE@Ni-DMF. A similar trend is observed for the *F1s* XPS spectra (Supplementary Fig. 37). Both LiF and anion species (−SO$_2$F) are uniformly distributed throughout the depth profiling in LPE-SEI, indicating the incomplete decomposition of −SO$_2$F[43]. In comparison, LPE@Ni-DMF-SEI exhibits a layered distribution of fluorine species with superficial −SO$_2$F and LiF inner phases. A LiF-rich inorganic layer has been reported to stabilize the Li metal SEI and promote interfacial Li$^+$ transport[44,45]. The *Ni2p* XPS spectrum (Supplementary Fig. 38) indicates that no Ni signal is present in the SEI composition, which is consistent with the EDS mapping results.

Cryogenic transmission electron microscopy (cryo-TEM) was utilized to further unveil the micro-structure and local chemical composition of the SEI. Although the distribution of C, N, O, F, and S elements is concentrated in both LPE-SEI and LPE@Ni-DMF-SEI, there are notable distinctions in the induced depositional morphology. LPE@Ni-DMF-SEI has a compact and thinner morphology compared to LPE-SEI, which exhibits an uneven deposition (Supplementary Figs. 39, 40). High-resolution TEM image and corresponding fast Fourier transformation (FFT) analysis confirm that the organic phases dominate LPE-SEI, with only a few crystalline Li$_2$O (111), LiOH (001), or LiF (200) observed through the major amorphous area (Fig. 4d)[42,46]. Opposite results were found for LPE@Ni-DMF-SEI which exhibits a large amount of inorganic LiF and Li$_2$O grains with little amorphous area (Fig. 4e). The difference in the SEI composition is further quantitatively distinguished by calculating the fluorine-to-carbon (F/C) elemental ratio (Fig. 4f). A much higher F/C ratio in LPE@Ni-DMF-SEI demonstrates that the integrated Ni-DMF complex can induce an anion-derived inorganic-rich SEI.

The dynamic evolution process of SEI and the corresponding Li$^+$ deposition behavior under cycling condition were illustrated by FEM simulations. We find that the thickness of LPE-SEI increases unevenly

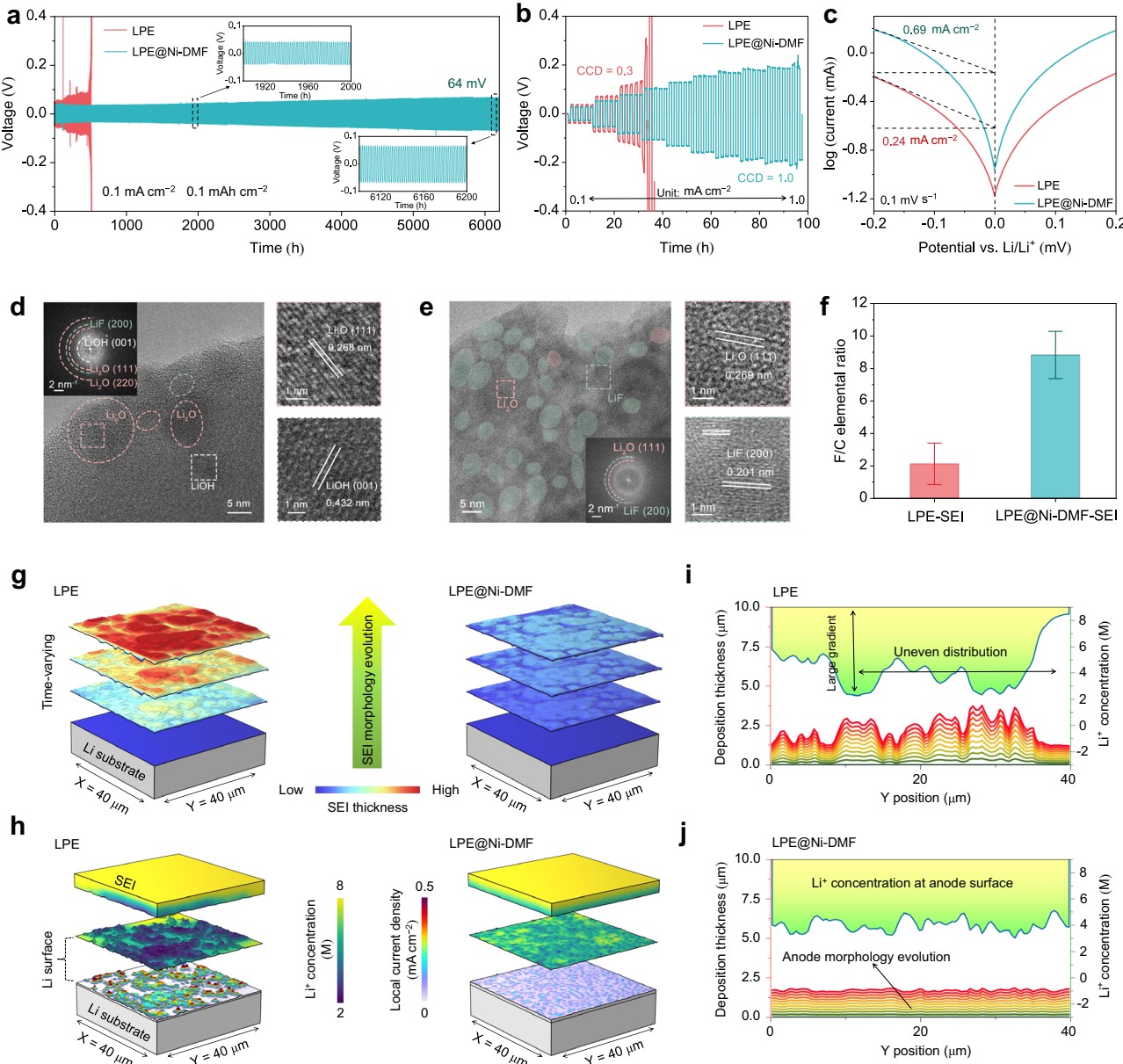

**Fig. 4 | Experimental study and FEM simulations on the interfacial stability of LPE@Ni-DMF with Li metal. a** Cycling stability of the Li||Li symmetric cells at a current density of 0.1 mA cm$^{-2}$ with capacity of 0.1 mAh cm$^{-2}$ per half cycle. The insets are the magnified images at the selected time period. **b** CCD of LPE and LPE@Ni-DMF. The plating/stripping time for half cycle is set to 1 h. **c** Tafel curves with calculated ECD of the Li||Li symmetric cells with a scanning rate of 0.1 mV s$^{-1}$. Cryo-TEM and corresponding FFT images for the component analyses in LPE-SEI (**d**) and LPE@Ni-DMF-SEI (**e**). **f** F/C elemental ratio in LPE-SEI and LPE@Ni-DMF-SEI. Values are means, and error bars were calculated by taking the standard errors from one sample for three measurements. **g** FEM simulation results of the evolution of SEI thickness in a Li||Li symmetric cell model. The color from blue to red represents the increase of SEI thickness. **h** The Li$^{+}$ concentration distribution at the interface between SEIs and Li anodes and the local current density distribution on Li surface (at the state of 1 h charging). The Li anode morphology evolution related to Li$^{+}$ concentration distribution (at the state of 1 h charging) on Li surface under the influence of (**i**) LPE-SEI and (**j**) LPE@Ni-DMF-SEI. The stacked curves from bottom to top represent the Li deposition thickness of different locations on the selected line, and the line colors change with the charging time.

with aggravated corrosion (Fig. 4g and Supplementary Movie 3; based on the DMF corrosion model), resulting in an uneven horizontal distribution of ionic conductivity of the SEI (Supplementary Fig. 41). In contrast, LPE@Ni-DMF-SEI exhibits a regulated morphology with a uniform ionic conduction behavior (Supplementary Movie 4). The SEI morphology after cycling obtained from the FEM model is in good agreement with the experimental observations from cryo-TEM. Considering that the Li deposition behaviors are highly correlated to the Li$^{+}$ concentration gradient in the SEI[47], we further compared Li$^{+}$ concentration distribution in LPE-SEI and LPE@Ni-DMF-SEI. Similarly, LPE-SEI exhibits evident local Li$^{+}$ depletion zones and extremely uneven Li$^{+}$

concentration distribution owing to its inhomogeneous SEI inducing sluggish Li$^{+}$ transport ability (Fig. 4h). Specifically, sharp Li$^{+}$ concentration gradient arises at local Li$^{+}$ depletion zones at the LPE-SEI–anode interface, which coincides with the areas with enlarged overpotential (Supplementary Fig. 42), causing excessively high local current density (Supplementary Fig. 43) and dendritic Li deposition (Fig. 4i, Supplementary Fig. 44 and Movie 5)[48]. As opposed to behavior of LPE, LPE@Ni-DMF yields the dense and smooth Li deposition due to the accelerated kinetics of Li$^{+}$ flux and uniformly distributed Li$^{+}$ concentration at the interface, which is further confirmed by SEM images (Fig. 4j, Supplementary Fig. 45 and Movie 6).

## Performance of Li | LPE@Ni-DMF | SPAN solid-state batteries

The favorable ion transport and interfacial features of LPE@Ni-DMF manifest its great promise in practical solid-state batteries. SPAN cathodes (high theoretical capacity of 1675 mAh g⁻¹) and Li metal anodes were paired for both LPE and LPE@Ni-DMF to assemble Li| LPE | SPAN and Li|LPE@Ni-DMF | SPAN coin cells and were tested at multiple temperatures. As shown in Fig. 5a, the rate performance (tested at room temperature) vastly improves for the Li|LPE@Ni-DMF | SPAN cell over Li|LPE | SPAN, delivering a specific capacity of 899 mAh g⁻¹ at 0.3 C, 872 mAh g⁻¹ at 0.5 C, and 811 mAh g⁻¹ at 1 C, respectively (here, all the coin cells were activated at 0.1 C for the first cycle). In terms of long-term cycling at room temperature, the Li| LPE@Ni-DMF | SPAN cell exhibits a reversible specific capacity of 961 mAh g⁻¹ (capacity retention of 81.9%) after 300 cycles at 0.2 C (Supplementary Fig. 46). By sharp contrast, the SPAN/LPE/Li cell exhibits rapid capacity decay during cycling, leading to a capacity retention of only 14.7% under the same conditions. Notably, the Li| LPE | SPAN cell suffers from severe capacity drop between 150 and 200

cycles accompanied by serious fluctuation of Coulombic efficiency (CE), which can be attributed to micro-short circuiting caused by the Li dendrites (Supplementary Fig. 47). The improved electrochemical performance of the Li|LPE@Ni-DMF | SPAN cell is also validated by cycling at 1 C (Fig. 5b). The Li|LPE@Ni-DMF | SPAN cell displays remarkable cyclability (1000 cycles with a capacity retention of 60%) and records an average CE of 99.8% throughout its cycle life. In contrast, the Li|LPE | SPAN cell fails after only 100 cycles with a low average CE of 96.7%.

In-situ electrochemical impedance spectroscopy (EIS) measurements and distribution of relaxation time (DRT) analysis were conducted to evaluate the Li⁺ transport behaviors during cycling. The impedance during the first 5 cycles increases in LPE, while the opposite trend is observed in the case of LPE@Ni-DMF (Supplementary Fig. 48). To further decouple the electrochemical processes, DRT technology is utilized to convert the impedance information from the frequency domain into the time domain[49]. The transformation allows the relatively objective and clear identification of the contribution of each

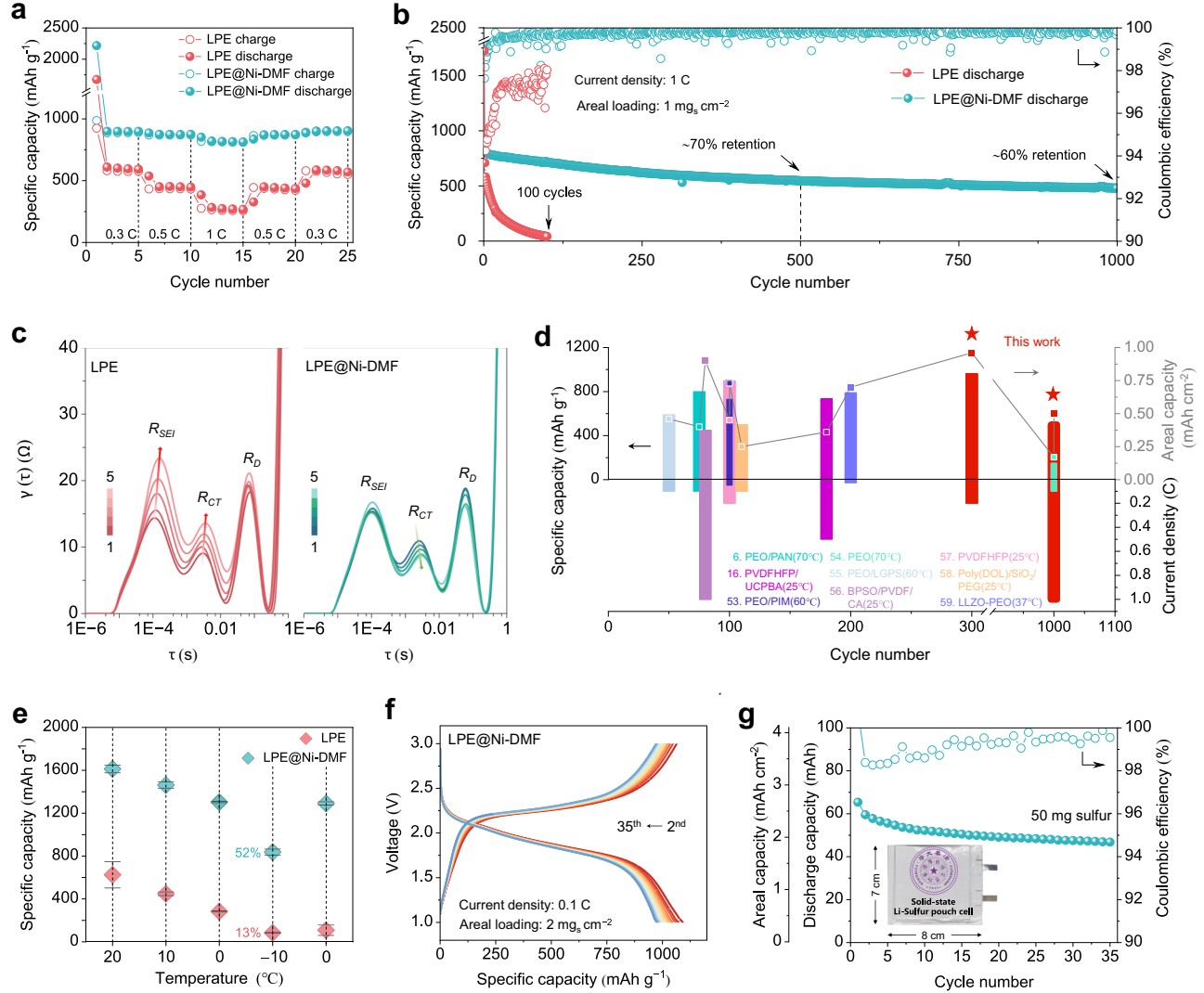

**Fig. 5 | Properties of the Li | |SPAN solid-state batteries using LPE@Ni-DMF.** **a** Rate performance of the Li | |SPAN coin cells. **b** Cycling stability of the Li | |SPAN coin cells at 1 C with a sulfur loading of 1 mg cm⁻². **c** DRT analyses of Li | |SPAN cells in the first 5 cycles. **d** Comprehensive comparison of polymer-based ssLSBs performance (including specific capacity, areal capacity, and cyclability) between this work and those in previous reports. **e** Temperature-dependent performance of Li | |SPAN coin cells at 0.05 C. Values are means, and error bars were calculated by

taking the standard errors from one sample for five measurements. **f** Charge-discharge profiles of the Li|LPE@Ni-DMF | SPAN pouch cell at 0.1 C with a sulfur loading of 2 mg cm⁻². **g** Cycling stability of the Li|LPE@Ni-DMF | SPAN pouch cell with 50 mg sulfur and capacity of 47 mAh. The inset is the showcase of the solid-state Li-Sulfur pouch cell. All the coin cells and pouch cells were tested in the voltage range from 1 to 3 V.

electrochemical process. As shown in Fig. 5c, both DRT spectra consist of three peaks, each with a unique relaxation time that denotes a specific electrochemical process. Specifically, the peaks located at time constants of $10^{-4}$, $10^{-3}$–$10^{-2}$, and $10^{-2}$–$10^{-1}$ s represent the impedances of SEI ($R_{SEI}$)[50], charge transfer ($R_{CT}$)[51], and diffusion process ($R_D$)[52] of the electrode, respectively. The impedance increases and right shift at $10^{-4}$ s in LPE can be attributed to the continuous growth of the organic-rich SEI layer with slower $Li^+$ transport kinetics. In contrast, the impedance in LPE@Ni-DMF remains invariant, demonstrating that the SEI layer with inorganic compounds as dominant contents is more stable once formed, which is essential for cyclability. We also observe impedance increase at $10^{-3}$–$10^{-2}$ s in LPE. The unstable SEI layer and contact loss is considered responsible for the hindered $Li^+$ charge transfer and sluggish electrochemical reduction kinetics. In the case of LPE@Ni-DMF, the impedance of charge transfer gradually decreases within the first 5 cycles, which could be attributed to the reinforced inorganic-rich SEI and accelerated $Li^+$ transport. The above results demonstrate that $Li^+$ transport regulation and DMF confinement by using Ni-DMF lead to notably improved cyclability, which is superior to previous reports (Fig. 5d)[53–59].

In addition to the excellent performance at room temperature, the Li|LPE@Ni-DMF|SPAN cell also shows exceptional performance at low temperatures. The temperature-dependent performance of Li|LPE|SPAN and Li|LPE@Ni-DMF|SPAN cells are compared at 0.05 C at 20 °C, 10 °C, 0 °C, and −10 °C (Supplementary Fig. 49). The average specific capacity of the Li|LPE@Ni-DMF|SPAN cell at −10 °C is as high as 835 mAh $g^{-1}$, corresponding to a capacity retention of 52% upon that at 20 °C, while the Li|LPE|SPAN cell only provides an average specific capacity of 86.5 mAh $g^{-1}$ at −10 °C, corresponding to a low capacity retention of 13% (Fig. 5e). Furthermore, the Li|LPE@Ni-DMF|SPAN cell is found to fully restore its capacity at 0 °C even after operation at −10 °C, suggesting its good reaction kinetics at low temperature. This can be attributed to the higher ionic conductivity ($2.55 \times 10^{-4}$ S $cm^{-1}$ at 0 °C and $1.26 \times 10^{-4}$ S $cm^{-1}$ at −10 °C; Supplementary Fig. 50) and better interfacial stability of LPE@Ni-DMF. Overall, the impedance of the Li|LPE@Ni-DMF|SPAN cell at 0 °C (465 Ω) is markedly lower than that of Li|LPE|SPAN cell (861 Ω; Supplementary Fig. 51).

Finally, to showcase the capability of LPE@Ni-DMF under more realistic conditions, a pouch cell with high areal loading (2 mg $cm^{-2}$) cathode was fabricated. The Li|LPE@Ni-DMF|SPAN pouch cell (50 mg sulfur loading) exhibits exceptional cycling performance at 0.1 C, delivering a high specific capacity of 950 mAh $g^{-1}$ after 35 cycles (Fig. 5f). The discharge-charge curves of Li|LPE@Ni-DMF|SPAN pouch cell are substantially overlapped from the 2nd to 35th cycles with negligible hysteresis aggravation, indicating excellent electrochemical reversibility. The pouch cell achieved a capacity of 47 mAh and areal capacity of 1.9 mAh $cm^{-2}$ after 35 cycles, demonstrating the feasibility of this CPE as a solid-state electrolyte for practical ssLSBs (Fig. 5g).

## Discussion

In this work, we propose a locally solvent-tethered polymer electrolyte design to achieve highly ion-conductive and durable CPEs with the dual-functional Ni-DMF additive as a proof of concept. Our results demonstrate that the confinement effect of Ni-DMF prevents DMF consumption, resulting in enhanced durability for both the CPE and Li metal anode (over 6000 h of cycling at 0.1 mA $cm^{-2}$ with capacity of 0.1 mAh $cm^{-2}$ per half cycle). Furthermore, Ni-DMF provides a locally DMF-rich interface for $Li^+$ transport, leading to improved ionic conductivities at both room temperature ($6.5 \times 10^{-4}$ S $cm^{-1}$) and extreme operating conditions ($2.55 \times 10^{-4}$ S $cm^{-1}$ at 0 °C and $1.26 \times 10^{-4}$ S $cm^{-1}$ at −10 °C). The Ni-DMF complex effectively addresses the Achilles' heel of solvent-containing SPEs, which is crucial for achieving stable cycling performance in ssLSBs (1000 cycles at 1 C). Notably, solid-state pouch cell with a high areal capacity of 1.9 mAh $cm^{-2}$ exhibited excellent

cyclability (47 mAh after 35 cycles). This work demonstrates a promising CPE design for the potential application of ssLSBs, and we expect that the design principle can be extended to SPEs for other emerging solid-state battery systems.

## Methods

### Materials preparation

$Ni(H_2O)_2Ni[CN]_4 \cdot xH_2O$ was synthesized via a coprecipitation method. In a typical procedure, 0.19 g of $NiCl_2 \cdot 6H_2O$ (purity: >99%, Macklin) and 0.176 g of dihydrate tri-sodium citrate (purity: >99%, Aldrich) were dissolved in 40 mL of deionized water under stirring for 1 h to form a clear solution. 0.193 g of $K_2[Ni(CN)_4]$ (AR, Aldrich) was added into 40 mL deionized water to form another clear solution. After that, the two solutions were mixed under agitated stirring for 4 h and then aged at 25 °C for 24 h. The light-blue precipitate was collected out and washed with deionized water three times denominated as $Ni-H_2O$. Followed by thermal dehydration treatment of $Ni-H_2O$ under a vacuum condition at 80 °C for 6 h, the light-yellow sample ($Ni(H_2O)_2Ni[CN]_4$) was achieved denominated as Ni-activated. 100 mg of Ni-activated powder was immersed in 2 mL DMF (purity: >99.8%, Aldrich) solution (~1.9 g), followed by magnetically stirring for 2 h to fulfill thorough solution exchange process. After the color of Ni-activated thoroughly changed to light-green, the residual DMF and $H_2O$ molecules were eliminated by centrifugation and the light-green precipitate ($Ni(DMF)_2Ni[CN]_4$) underwent vacuum drying at 60 °C for 2 h to yield the final product Ni-DMF.

The solid-state electrolyte (SSE) membranes were prepared through a solution-casting method. 0.3 g of PVDF-HFP (M.w. ~400,000, Maclin) was dispersed in 1.5 mL of DMF solution (~1.42 g) at 80 °C for 2 h. 0.3 g of LiFSI (purity: >99.9%, DoDoChem) was dissolved in 1.5 mL of DMF solution (~1.42 g) at 25 °C for 1 h. PVDF-HFP and LiFSI solutions were then mixed, followed by stirring at 25 °C for 4 h. Then the mixed solution was cast onto a glass substrate using a doctor blade with 750 μm height, and the DMF solution was evaporated at 60 °C for 4 h to obtain the LPE membrane. With other conditions unchanged, 60 mg of Ni-DMF powder was added into the mixture of PVDF-HFP and LiFSI solution with another 2 h mixing performed before the casting process (the weight ratios between Ni-DMF, PVDF-HFP, LiFSI, and DMF is measured to be 1:5:5:47), and the LPE@Ni-DMF membrane can be obtained.

The SPAN composite was prepared by mixing sulfur (Alfa Aesar) and polyacrylonitrile (Mw = 150,000, Aldrich) in a weight ratio of 4:1, followed by heating treatment in an Ar-filled furnace at 450 °C for 6 h[16]. The cathode slurry was firstly prepared by heat-dissolving 0.2 g of PVDF-HFP and 0.2 g of LiFSI in 5 mL DMF solution at 80 °C for 4 h. Then, 0.5 g of SPAN and 0.1 g of multi-walled carbon nanotubes (>95%, XFNANO) were mixed into the slurry. The electrodes were prepared by coating the slurry onto an Al foil (15 μm, Canrd) and dried at 60 °C for 12 h. The sulfur loading of SPAN cathodes was calculated to be 1–2 mg $cm^{-2}$. All the preparation processes were carried out in an Ar-filled glovebox.

### Electrochemical measurements

The as-prepared LPE and LPE@Ni-DMF membranes were cut into disks with different diameters for electrochemical measurements. Stainless steel (SS) symmetric SS|SSE|SS cells were used to test ionic conductivities at 25 °C, 20 °C, 10 °C, 0 °C, and −10 °C, respectively, according to the equation of:

$$\sigma = L/(R \times S) \qquad (2)$$

where L, R, and S are the thickness, bulk resistance, and area of SSE (with a diameter of 15 mm), respectively, using a Biologic multi-channel electrochemical workstation in the range from high frequency of 100 K Hz to low frequency of 0.1 Hz and an amplitude voltage of

5 mV. The $E_a$ was calculated according to the Arrhenius equation of:

$$K = Ae^{-E_a/RT} \tag{3}$$

where K is rate constant, A is pre-exponential factor, R is gas constant, and T is absolute temperature.

Symmetric Li|SSE|Li cells were used to test $t_{Li^+}$ by combining AC impedance and potentiostatic polarization procedures according to the equation[5]:

$$t_{Li^+} = \frac{I_S(\Delta V - I_0 R_0)}{I_0(\Delta V - I_S R_S)} \tag{4}$$

Where $\Delta V$ is the DC polarization voltage (10 mV), $I_0$ and $I_s$ are initial and stable currents (μA) during polarization, and $R_0$ and $R_s$ are the impedance (Ω) before and after polarization.

Cyclic voltammetry (CV) measurements of the Li|SSE|Li cells were performed on an electrochemical workstation at initial open-circuit voltage over the voltage range of −0.2−0.2 V with a scanning rate of $0.1 \, mV \, s^{-1}$. Tafel curves were obtained by fitting the second cycle (after SEI formation) of CV data including anodic scan and cathodic scan. The voltage at the intersection of two curves was defined as 0 V (around the Li/Li⁺ equilibrium potential). The electrochemical window of SSE was obtained at 25 °C by linear sweep voltammetry (LSV) using asymmetric Li|SSE | SS cells at 0−5 V (vs. Li/Li⁺) with a scanning rate of $1 \, mV \, s^{-1}$.

DRT analysis is conducted using the DRT Tools which are developed by Ciucci. Group (GitHub-ciucaslab/DRTtools: An intuitive MATLAB GUI to compute the DRT). The transition from frequency domain to time domain is accomplished by the general formula:

$$Z(\omega) = R_\infty + \int_0^\infty \frac{\gamma(\tau)}{1 + j\omega\tau} d\tau \tag{5}$$

Where $Z(\omega)$ represents the impedance based on frequency, $R_\infty$ represents ohmic impedance in the battery, $\tau$ represents the specific relaxation time, and $\gamma(\tau)$ is the distribution function of relaxation times.

A LAND CT2003A electrochemical testing system was used to measure the electrochemical performance of symmetrical and asymmetric coin cells at multiple temperatures. The SSE membranes were cut into disks with diameter of 19 mm for tests. The cycling performance of the Li|SSE|Li cells was tested at current densities of $0.1−1 \, mA \, cm^{-2}$ at 25 °C. The performance of Li|SSE | SPAN cells was tested with the voltage range from 1.0 to 3.0 V at 25 °C, 20 °C, 10 °C, 0 °C, and −10 °C, respectively. For the pouch cells assembly, SPAN coated on the Al current collector was used as the cathode (areal loading: $2 \, mg_s \, cm^{-2}$), and lithium foil (100 μm) on the Cu current collector (9 μm, Canrd) was used as the anode. Both the anode and cathode were $5 \times 5 \, cm^2$ while the solid-state electrolyte was $6 \times 6 \, cm^2$. It is noted that the Al current collector was welded to an Al tab (Canrd) and the Cu current collector was welded to a Ni tab (Canrd) with tab sizes of $0.1 \times 5 \, cm^2$. The whole electric core was finally packed with Al-plastic film (113 μm, Canrd) in a drying room with a humidity of less than 10% at 25 °C.

## Materials characterization
The morphologies and microstructures of samples were characterized by SEM (5 kV, Hitachi SU8010) with an EDS detector and TEM (JEOL JEM-2100f). The DMF corrosion procedure on Li metal was recorded by optical microscope (DVM6). XRD (Bruker AXS D8) and ATR-FTIR (Thermo Nicolet Avatar 320 Spectrometer) spectra were employed for analysis of the crystal structures and composition of samples. TG (TA instruments Q500) analysis was conducted to probe the thermal stability of SSE and determine the weight variation of DMF before and after electrochemical tests. Ni K-edge XANES measurements and EXAFS analyses were performed at Canadian Light Source (CLS), using the soft X-ray Micro-characterization beamline (SXRMB, 06B1-1). The STXM O K-edge and Ni L-edge measurements were conducted using the SM beamline (10ID-1) at the CLS. The ⁷Li ssNMR spectra of samples were recorded on a 500 MHz Bruker AVANCE NEO spectrometer equipped with an 11.70 T widebore magnet using a 2.5 mm Bruker MAS probe (DVT design) at 25 kHz MAS. The inner structures and components reconstruction of SSE membrane were executed by X-CT (Zeiss Xradia 515 versa). The mechanical performance was tested by a film stress measurement system (FSM 500TC-R). The Raman spectra were obtained by LabRAM HR Evolution to detect the FSI⁻ anion states. The in-situ reactivity between DMF and Li metal was investigated using an optical microscope (DVM6). The lithium foil was cut to a size of $0.5 \times 1 \, cm^2$ and then placed in the porthole of a Teflon mold. The porthole was then filled with DMF solvent and sealed with a glass slide. All operations were performed in an Ar-filled glovebox. Two-dimensional wide-angle X-rays scattering (2D-WAXS) tests were conducted on Brockhouse X-ray Diffraction and Scattering Beamlines-High Energy Wiggler (BXDS-WHE) Beamline at CLS. For XPS measurements, each Li tablet (after Li//Li cell cycling) was washed with 1,2-Dimethoxyethane (purity: >99.9%, DoDoChem) solvent for three times to remove the residual DMF on Li tablet. Then the washed Li tablet was dried under vacuum for 6 h at 25 °C before transferred and sealed into the XPS holder in the Ar-filled glovebox. The XPS profiles were collected with an Escalab Xi+ scanning XPS microprobe. Cryo-TEM characterizations were performed on TEM (FEI Talos-S) operated at 200 kV with a Gatan 698 cryo-transfer holder. In the cryo-TEM test, 300 mesh Cu as a carrier, and the samples were kept stable by liquid nitrogen freezing at the low temperature of −180 °C.

## Finite element method simulation
Based on the COMSOL Multiphysics 6.0 platform, we performed the finite element analysis for the process of DMF corrodes the Li anode and the Li⁺ electrochemical behavior under different SEIs, and the following are the simulation equations:

## Mass transfer
In the electrolyte, the transfer of ions is driven by migration because of electric field and diffusion because of concentration gradient which are governed by the Nernst−Planck equation:

$$N_i = -D_i\left(\nabla c_{0,i} - \frac{z_i F c_{0,i}}{RT}\nabla\varphi_l\right) \tag{6}$$

Where $N_i$ is flux, and $D_i$, $z_i$ and $c_{0,i}$ is the diffusion coefficient, charge and concentration of species i, respectively. F is the Faraday's constant, R is the ideal gas constant, T is the Kelvin temperature and $\varphi_l$ is the electrolyte potential. Meanwhile, the ions in the electrolyte follows the equations of conservation of mass and charge:

$$\frac{\partial c_i}{\partial t} + \nabla \times N_i = 0 \tag{7}$$

$$\sum_i z_i c_i = 0 \tag{8}$$

Where $c_i$ is the concentration, and $z_i$ is the valence of each species in the electrolyte.

## Corrosion reaction of DMF with Li anode
The corrosion reaction between DMF and Li at the anode surface could be assumed as:

$$DMF + Li^+ + e^- \leftrightarrow LiA + B \tag{9}$$

Where LiA and B are the corrosion products, which may contain a number of inorganic components according to XPS results. We artificially defined that the DMF could be transferred together with Li$^+$ in LPE, while in LPE@Ni-DMF, the mobility of DMF would be set to 0 to simulate the confine effect.

In order to simulate the random corrosion results, we set a random distribution function at the electrode-electrolyte interface, the function is generated by:

$$f(x,y) = \sum_{m=-M}^{M} \sum_{n=-N}^{N} a(m,n) \cos[2\pi(mx + ny) + \theta(m,n)] \quad (10)$$

Where x and y are the spatial coordinates, m and n are the spatial frequencies, a(m,n) is the amplitude, and θ(m,n) is the phase angle. The amplitude is randomly generated by a Gaussian distribution function, and the phase angle and spatial frequency are derived from a uniform random distribution in a limited interval.

## Kinetics of Li$^+$

Basically, Li$^+$ is transported from the bulk solution to the anode surface then reduced to Li-atom, and at the interface of the electrolyte and the anode, the deposition process of Li$^+$ can be described as the simplified reaction:

$$Li^+ + e^- \leftrightarrow Li \quad (11)$$

The electrochemical behaviors of Li$^+$ at the electrode-electrolyte interface could be described by the famous Butler-Volmer equation:

$$i_{loc} = i_{ex} \left[ \exp\left(\frac{\alpha_a F\eta}{RT}\right) - \exp\left(\frac{-\alpha_c F\eta}{RT}\right) \right] \quad (12)$$

Where $i_{loc}$ is the local current density, which can be employed to quantify the local reaction rate. η is overpotential, while $\alpha_a$ and $\alpha_c$ are the anodic and cathodic charge transfer coefficients, respectively. The exchange current density ($i_{ex}$) is defined as the current density flowing equally in each direction at the reversible potential. It is a useful metric for characterizing the ease of a reaction to occur, and is closely related to the electron transfer kinetics and the concentration gradient near the surface:

$$i_{ex} = i_e \prod_{i,v_j>0} \left(\frac{c_{Li^+}}{c_b}\right)^{\frac{\alpha_a v_j}{n_j}} \prod_{i,v_j<0} \left(\frac{c_{Li^+}}{c_b}\right)^{\frac{-\alpha_c v_j}{n_j}} \quad (13)$$

Therefore, the $i_{ex}$ is significantly influenced by the $\frac{c_{Li^+}}{c_b}$, which is also refers to the concentration gradient. $c_{Li^+}$ is the concentration of Li$^+$ near the anode and $c_b$ can be considered as the concentration of Li$^+$ in the bulk electrolyte. The current density ($i_e$) represents the kinetics of electrons. $v_j$ is the stoichiometric coefficients and $n_j$ is the number of electrons transferred. The overpotential (η) can be decided into the concentration overpotential ($\eta_c$) and the electrochemical overpotential ($\eta_e$), which can be calculated from:

$$\eta = \varphi_s - \varphi_l - U_{eq} = \eta_c + \eta_e \quad (14)$$

Where $\varphi_s$ is the solid phase potential, and $U_{eq}$ is the equilibrium potential of the reaction.

Based on the above equations, the $i_{loc}$ is closely related to the Li$^+$ concentration distribution and the overpotential at the anode surface. Therefore, boundary conditions near the substrate can be described as:

$$N_{Li^+} \cdot \mathbf{n} = -\frac{i_0}{2F} \left[ \exp\left(\frac{\alpha_a F\eta}{RT}\right) - \frac{c_{Li^+}}{c_0} \exp\left(\frac{\alpha_c F\eta}{RT}\right) \right] \quad (15)$$

Where **n** is the normal vector of the boundary.

## Morphology evolution of SEI, corrosion and Li$^+$ deposition

We use the Dynamic Mesh method to simulate the corrosion and Li$^+$ deposition morphology, and the Level Set method to keep track of SEI morphology evolution. The Level Set interface automatically sets up the equations for the movement of the interface between the electrolyte and electrode domains. The interface is represented by the 0.5 contour of the level set variable. The level set variable Φ varies from 1 in the electrode domain to 0 in the electrolyte domain. The transport of the level set variable is given by:

$$\frac{\partial \Phi}{\partial t} + \mathbf{u}\nabla\Phi = \gamma\nabla \cdot \left( \varepsilon\nabla\Phi - \Phi(1-\Phi)\frac{\nabla\Phi}{abs(\nabla\Phi)} \right) \quad (16)$$

The ε parameter determines the thickness of the interface and is defined as $\varepsilon = h_{max}/4$, where $h_{max}$ is the maximum mesh element size in the domain, and **u** is the velocity of level set. The γ parameter determines the amount of reinitialization. A suitable value for γ is the maximum velocity magnitude occurring in the model.

The Level Set delta function δ to prescribe the growth of SEI is approximated by:

$$\delta = 6abs(\Phi(1-\Phi)) \cdot abs(\nabla\Phi) \quad (17)$$

It should be noted that in order to improve the convergence of numerical calculations and clarify the role of each step, the impact of DMF corrosion Li anode related to the formation of SEI and the electrochemical behavior of Li$^+$ under different SEIs were calculated in two steps. Firstly, the height information of SEI generated by DMF corrosion on Li anode is extracted as the basis for subsequent simulation of SEI electrochemical properties. Secondly, Li$^+$ deposition is further simulated based on the aforementioned SEI properties. Both the anodic and cathodic charge transfer coefficients are set as 0.5, the exchange current density is set as 100 A m$^{-2}$, the Li$^+$ concentration in LPE is set as 8 M, the temperature is fixed at 298 K, and the applied current density is 1 mA cm$^{-2}$. The diffusion coefficient of Li$^+$ in the electrolyte is set to 1e$^{-11}$ m$^2$ s$^{-1}$.

## Calculations

Classical molecular dynamics (MD) simulations were performed to obtain the ionic conduction mechanism of the LPE and LPE@Ni-DMF with the Large Scale Atomic/Molecular Massively Parallel Simulator (LAMMPS) software package[60]. The nonpolarizable UFF[61] was utilized to parameterize the bonded and van der Waals interactions of Ni-DMF, which was generated using LAMMPS interface[62]. The bonded and nonbonded parameters for DMF were obtained from the Optimized Potentials for Liquid Simulations All Atom (OPLS-AA) force fields[63,64], while those for Li$^+$ and FSI$^-$ are taken from ref. 65 and ref. 66. Partial charges of the Ni-DMF were fitted from first principles using the RESP2 method[67] with B3LYP-D3(BJ)/def2TZVP[68] level of theory for Ni and CN ligand. Long-range electrostatic interactions were handled by the particle-particle particle-mesh (PPPM) solver with a grid spacing of 0.1 nm. A cutoff distance of 1.25 nm was used for electrostatic and 12−6 Lennard-Jones interactions. The fitted partial charges of Ni-DMF and the unit charge of Li$^+$ (+1) and FSI$^-$ (−1) were then scaled by a factor of ε = 0.6 and 0.8 for LPE@Ni-DMF and LPE, respectively, to account for the fact that solvent−ion and ion−ion interactions are typically overestimated in nonpolarizable force fields[69–71].

The atomic position obtained from single crystal X-ray diffraction of LPE@Ni-DMF was used as the initial structure to build a supercell containing 128 Ni, 128 CN, 512 DMF, and 180 LiFSI. LPE was modeled with 192 DMF, 264 LiFSI to obtain an effective concentration of 7.15 M, which agrees to the experimental concentration of 8.3 M.

The initial configurations were minimized by conjugated-gradient energy minimization scheme employing a convergence criterion of $1.0 \times 10^{-4}$. The LPE and LPE@Ni-DMF structures were then equilibrated for 2 ns in the isothermal-isobaric ensemble (T = 298, 283, 263 K, and P = 1 bar) using the Parrinello–Rahman barostat followed with an annealing process. Subsequently, 50 ns production runs within the canonical ensemble (NVT) under a Nosé–Hoover thermostat were performed to obtain the $Li^+$ transport properties. The simulation time was long enough to sample adequately the Fickian (diffusive) regime of all systems[72].

The $H_2O$ and DMF exchange free energy, highest occupied molecular orbital (HOMO), and adsorption energy were calculated using B3LYP-D3(BJ) def2TZVP level of theory with implicit solvent model IEF-PCM (UFF, DMF)[73,74] in Gaussian 16 (ref. [75]). Quasi-harmonic entropy and enthalpy correction with a cutoff frequency of 100 $cm^{-1}$ was applied as suggested by ref. [76]. using the GoodVibes program[77].

## Data availability
All data supporting the research in this paper are available in the main text and Supplementary Information, and source data can be obtained through reasonable requests from corresponding authors.

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

## Acknowledgements

This work was supported by the National Key Research and Development Program of China (2021YFB2500200), National Natural Science Foundation of China (No. 52072205), Joint Funds of the National Natural Science Foundation of China (U21A20174), Guangdong Innovative and Entrepreneurial Research Team Program (2021ZT09L197), Shenzhen Science and Technology Program (KQTD20210811090112002), and the Start-up Funds, the Overseas Research Cooperation Fund of Tsinghua Shenzhen International Graduate School and the Shenzhen Geim Graphene Center. The authors thank J.Wang. who helped us with the Scanning Transmission X-ray Microscopy tests on SM Beamline at CLS. The CLS is supported by the NSERC, the National Research Council Canada, the Canadian Institutes of Health Research, the Province of Saskatchewan, Western Economic Diversification Canada, and the University of Saskatchewan.

## Author contributions

Y.Z. and G.Z. conceived the idea. G.Z. and T.H. directed the project. Y.Z. and Z.L. synthesized materials, characterized materials, and analyzed data. T.H., X.W., and C.J. performed DFT calculations and molecular dynamics simulations. M.Z. performed FEM simulations and data analysis. G.L. performed cryo-TEM and cryo-TEM EDS experiments. J.W. conducted the XAS and STXM experiments. X.X., Z.H., R.G., L.N., and Y.S. helped with electrochemical measurements and battery test. Y.Z., Z.P., and G.Z. cowrote and revised the manuscript. All authors contributed to the interpretation of the results.

## Competing interests

The authors declare no competing interests.
