## [Peer Review File · Nature Communications]

A locally solvent-tethered polymer electrolyte for long-life lithium metal batteriesREVIEWER COMMENTS

Reviewer #1 (Remarks to the Author):

This research aimed to address the issues related to the low ionic conductivity of solid polymer electrolytes and the instability of lithium-metal in DMF with high polarity. It tackled these challenges by utilizing the ligand-assisted transport mechanism and conducted extensive simulations to provide insights into these solutions. However, there are some unconvincing parts in terms of interpreting the experimental results. It seems that an ambiguous explanation is being made to connect the experimental results. Several inherent concerns were unavoidable before supporting this work for publication.

1. Looking at the simulation data, it seems that it takes a relatively long time for DMF to be included in the Li ion first solvation structure(0.76ns) and then return to the Ni-DMF crystal structure(36.79ns). If DMF does not bind well for a long period of time, it feels like there is enough time for DMF diffusion to occur. What do you think about this?

2-1 You mentioned the incomplete reduction product of SO₂F in the SEI component. What do you think is the difference between your electrolyte and the control group? Because FSI is strongly captured and fixed by Ni²⁺, it is thought that it will be difficult for anions to reach the electrode surface.

2-2 In general, in liquid electrolytes, a high transference number indicates that more anions are included in the lithium-ion first solvation structure. However, your paper states the opposite. What do you think about the relationship between the lithium hopping mechanism and inorganic-based SEI components?

2-3 Could you provide the pick intensity of XPS data?

2-4 And how can the charge imbalance between the two electrodes resulting from immobilization of anions be interpreted?

3. It was said that DMF reactivity and low ionic conductivity were solved simultaneously, but most of the data was interpreted based on simulation. Can the above results (Especially, Reduction intensity of solvent decomposition peak in CV.) be obtained through electrochemical experiment?

Reviewer #2 (Remarks to the Author):

This is a beautiful mechanistic study of how Ni bound DMF can increase the ionic conductivity of polymer electrolytes and at the same time not to be consumed by reaction with Li anode. The study has many complimentary tools such as TGA, SAXS/WAXS, TEM and simulation to substantiate their conclusions. This work can be published as is. I just have a minor question - how do the authors determine how much Ni-DMF to incorporate into the LPE? Does the starting LPE and CPE have similar DMF content?

Reviewer #3 (Remarks to the Author):

Zhu et al. created a Hofmann-DMF coordination complex and incorporated it into a PVDF-HFP matrix to obtain a composite-polymer-electrolyte (CPE) membrane labeled as LPE@Ni-DMF for solid-state batteries. The LPE@Ni-DMF showed the ionic conductivity of 6.5×10^{-4} S cm⁻¹ at room temperature, which is an acceptable level given the classification of solid electrolytes (i.e., CPEs), and enabled stable cycling of the Li|Li symmetric cell at 0.1 mA

cm⁻¹, 0.1 mAh cm⁻¹ (=2 h/cycle) for ≥3,000 cycles. The authors deduced the origins of the long cycling stability of LPE@Ni-DMF are improved Li⁺ conductivity and uniformly distributed Li⁺ ions at the Li|solid-electrolyte interface, which allowed the smooth deposition of Li metal. Their experimental and computational data support this consideration. They also demonstrated the applicability of this solid-electrolyte membrane to solid-state lithium-sulfur batteries using a sulfurized polyacrylonitrile (SPAN) cathode with a sulfur loading of 1 or 2 mg cm⁻² in both coin-cell and pouch-cell formats, of which the coin cell maintained a specific capacity of ≥500 mAh/g (i.e., ≥0.5 mAh/cm²) for ≥1,000 cycles (at room temperature, 1C).

Overall, the results look great; Ample experiments using various techniques have been conducted from material characterization to performance evaluation and its reasoning, which will definitely be informative for researchers in the battery field.

However, at this stage, I hesitate to provide a positive recommendation for this manuscript given (a-1) the hitherto progress of the research field, (a-2) the mismatch between the focused issue in the manuscript and the authors' approach, (a-3) unknown electrochemical stability of the authors' Hofmann frameworks, (a-4) undisclosed issue associated with Ni species coming from the authors' Hofmann frameworks, and (a-5) unspecified effect of Ni-DMF addition on the interfacial properties of the LPE. These concerns are shown in detail as follows:

[a. Major Concerns]

(a-1) Is the residual DMF in a polymer matrix really a serious problem?

In the Introduction section, the authors point out the focused issue in the manuscript: DMF decomposition on Li by saying "The uncontrolled decomposition and continuous depletion of DMF are highly detrimental to the stability of the Li metal anode as well as the durability of SPEs." (Lines 62-65) After that, they try to signify the importance of their work by stating: "In brief, previous additive engineering approaches mainly focus on tackling one issue of either ionic conductivity or electrochemical stability. An effective strategy to simultaneously improve both aspects needs to be developed for the ultimate application of SPEs." (Lines 81-85)

However, based on the provided references and information from other papers, I could not justify the decomposition of DMF trapped in a polymer matrix as a major obstacle to the solid-state-battery application of SPEs.

(a-1-1) For instance, in Ref. 9, by which the authors claim that DMF decomposition on Li is serious (Lines 60-62), the opposite opinion can be found, where the authors of Ref. 9 say: "The strong [DMF-Li⁺] complex in the PVDF-DMF-LiFSI film eliminates the free DMF that can attack the metallic lithium. Moreover, it was reported that the stable solid electrolyte interphase (SEI) layer between the PVDF-DMF-LiFSI film and the lithium metal can suppress the lithium dendrite growth" with the 38 quoted references therein. Indeed, they show the stable Nyquist plot of the Li|PVDF-DMF-LiFSI|Li symmetric cell for at least 216 h and the stable cycling of a solid-state lithium-metal battery (i.e., Li|PVDF-DMF-LiFSI|LiNi_{1/3}Co_{1/3}Mo_{1/3}O₂) with no sign of capacity decay for at least 200 cycles at room temperature, 0.2C. Therefore, I am reluctant to say DMF decomposition needs to be addressed urgently.

(a-1-2) In addition, as for the CPEs containing dielectric ceramic materials, the authors state "However, due to the limited amount and surface area of additives, their ability to immobilize DMF, particularly during calendar aging, is insufficient." (Lines 79-81) by

mentioning Refs 13–15. However, these references, especially Refs 14 and 15 do demonstrate an exceptionally long cycling performance for both symmetric and asymmetric cells (=lithium-metal batteries). For the cycle stability of Li|Li symmetric cells, Ref. 14 shows $\geq 1,300$ cycles at 0.1 mA cm^{-1} , 0.1 mAh cm^{-1} (=2 h/cycle), while Ref. 15 shows ≥ 950 cycles at 0.1 mA cm^{-1} , 0.1 mAh cm^{-1} (=2 h/cycle), $25 \text{ }^\circ\text{C}$. As for the full cells, Ref. 14 shows the long-term cycling of Li|LiNi_{0.8}Co_{0.1}Mo_{0.1}O₂ with a capacity of $\geq 100 \text{ mAh/g}$ (i.e., $\geq 0.16 \text{ mAh/cm}^2$) for $\geq 1,500$ cycles at 2C, $25 \text{ }^\circ\text{C}$, while Ref. 15 shows the Li|LiNi_{0.8}Co_{0.1}Mo_{0.1}O₂ cell that can be cycled at 1C, $25 \text{ }^\circ\text{C}$ for $\geq 1,500$ cycles with a capacity of $\geq 100 \text{ mAh/g}$ (i.e., $\geq 0.16 \text{ mAh/cm}^2$). The difference in cell performance between them and the authors' results is less obvious and, therefore, it can be deduced that a continuous DMF decomposition on the Li side would not be detrimental to the long-term stability of SPEs.

(a-1-3) Furthermore, a vast number of past studies on the CPEs using metal-organic frameworks, to which the authors' current work might belong, have already exhibited excellent ionic conductivities of such CPEs and competitive symmetric and asymmetric cell performances, discoloring the significance of the authors' results to some extent. Please take a close look at the following review papers in this field and references therein (please cite them if relevant to do so in the revision stage):

- Interdisciplinary Materials 2 (2023) 475–510
- Electrochemistry Communications 150 (2023) 107491
- EcoMat 5 (2023) e12283

(a-1-4) It seems the authors' concept (i.e., the use of open metal sites of a framework to trap solvent molecules, thereby mitigating the issue associated with the solvent decomposition on Li) has been already proven by this past study: Journal of Power Sources 240 (2013) 653–658, where Zn₄O(BDC)₃ (BDC stands for 1,4-benzenedicarboxylate) was incorporated in a poly(ethylene oxide) (PEO) matrix with residual DMF molecules, and the main cause of the improved reversibility of the Li plating/stripping behavior and the improved rate and cycle performances of Li|LiFePO₄ solid-state cells were concluded as the result of the strong solvent-trapping ability of Zn₄O(BDC)₃. Although the used materials in this past study are different from the authors' work, it has demonstrated the same concept. Therefore, the authors would need to provide other evidence to clarify the originality of the work.

(a-2) Does mixing Ni-DMF with the LPE really help mitigate the decomposition of DMF molecules that are in the LPE matrix?

(a-2-1) Based on Figs. 3a and 3b, and Supplementary Videos S1 and S2, readers can identify that LPE@Ni-DMF has Ni-DMF grains that are fully covered with the LPE and the LPE forms 3D ion-conduction network throughout LPE@Ni-DMF. This means that the incorporation of Ni-DMF into the LPE cannot prevent the LPE from being exposed to the Li metal side. Therefore, regardless of Ni-DMF incorporation, the LPE that contains residual DMF molecules in its matrix definitely attaches to Li metal. In this context, the authors' results cannot be direct proof of the mitigated decomposition of DMF molecules in the LPE matrix using Ni-DMF.

Therefore, the authors' approach appears to be doing the same thing as additive engineering where the effect of DMF decomposition on cycle stability was minimized because the concentration of trapped DMF in a polymer matrix per unit volume was diluted. This dilution effect is also valid in the authors' case; Even though DMF molecules in Ni-DMF are immobilized greatly, those trapped in the PVDF-HFP matrix have no bearing on immobilization. Hence, the stable cycling of Li|Li symmetric cell using LPE@Ni-DMF appears

to benefit from the diluted amount of trapped DMF molecules in a polymer matrix in a unit volume, and, therefore, the authors' approach does not like the significant breakthrough to mitigate the adverse effect stemming from the decomposition of DMF molecules trapped in a polymer matrix.

(a-2-2) Additional information that would be helpful to support the authors' claim

- If Ni-DMF can absorb extra DMF molecules in its crystalline structure (shown in Fig. 1a) or not

- o If yes, please state this in the manuscript with additional information about the crystalline structure of Ni-DMF with excess DMF molecules (somewhat similar to Supplementary Fig. 1a)

- Regardless of the answer to the above comment, the authors would need to improve the preparation method of LPE@Ni-DMF because it does not look like a straightforward method to evaluate the solvent-trapping ability of Ni-DMF when mixed with the DMF-containing LPE. The issue is that all ingredients were mixed in DMF, which might have allowed solvent DMF molecules to fill up all the adsorption sites in Ni-DMF. Therefore, the DMF content in the LPE matrix in LPE@Ni-DMF and that in the reference LPE (per unit volume of the LPE) would be the same as each other. To highlight the significance of the authors' Hofmann frameworks in mitigating DMF decomposition on Li, the following experiments are recommended:

- o Mixing a melted LPE (<150 °C so as not to remove residual DMF molecules in the matrix) with Ni-DMF or Ni-activated and coating the mixture to obtain LPE@Ni-DMF or LPE@Ni-activated without any concerns about the presence of excess DMF molecules in the Hofmann framework

- o Inserting a Ni-DMF or Ni-activated thin layer between the LPE and Li and performing cell tests. The Ni-DMF or Ni-activated thin layer might be obtained by vacuum filtration of a Ni-DMF- or Ni-activated-containing solution or drop-casting of a Ni-DMF- or Ni-activated-containing solution on Li or the LPE followed by solvent evaporation

(a-3) What are the electrochemical potential windows of Ni-H₂O, Ni-activated, Ni-DMF, LPE, and LPE@Ni-DMF?

The electrochemical potential window is one of the important pieces of information about solid-electrolyte membranes, which allows battery scientists to decide the working potential range of batteries, but no such information has been described in the manuscript. It seems the authors have tried to include this in the manuscript based on the description for an unused linear sweep voltammetry (Lines 597–599). It would be nice if the authors could include this information in the manuscript.

(a-4) How do the Ni species in the authors' Hofmann frameworks affect the electrochemical reactions on the Li side (as well as the cathode side)?

It has been known that the transition-metal ions adversely affect the electrochemical reactions on the anode side [Adv. Energy Mater. 11 (2021) 2103005]. For example, transition metals of cathode active materials (e.g., LiNi_xCo_yMn_zO₂) would dissolve in an electrolyte solution (as cation forms), migrate to the anode, and participate in electrochemical reactions there. Reduced transition-metal ions would form metal nanoparticles on the anode, which acts as catalysts to promote lithium dendrite growth. With this in mind, it is of crucial importance to investigate whether Ni species in the authors' Hofmann frameworks adversely affect their cell performances or not. Since the Ni sources, i.e., NiCl₂·6H₂O and K₂[Ni(CN)₄], are originally small complexes that can dissolve in a solvent and the distance between the closest Ni-DMF in LPE@Ni-DMF and the Li-anode surface in the authors' experiments is not long, I believe there is a high likelihood in observing some sort of morphological, spectroscopical, and/or electrochemical footprints

stemming from the Ni species in Ni-DMF.

Some recommended experiments are SEM, EDS, and XPS of the Li surface after symmetric or asymmetric cell cycling to see if Ni species can be detected. If not detected, such results should be described in the manuscript to relieve the above concern.

(a-5) How does the incorporation of Ni-DMF into the LPE alter the ion-conduction behavior of the interface between Ni-DMF and the LPE?

In the manuscript, it seems not enough information has been stated for this point (except for Supplementary Fig. 26). However, many reports state the reduction of the crystallinity of a polymer attached to the surface of a material doped in its matrix, thereby improving the ionic conductivity of the composite material [e.g., Ref. 13, *Energy Environ. Sci.* 13 (2020) 2386–2403]. Importantly, some dielectric ceramic materials are known to induce the dehydrofluorination of PVDF as evidenced by the formation of C=C double bonds (e.g., Ref. 13), which alters the ion-conduction behavior at the interface between the doped material and PVDF.

Therefore, the authors are requested to provide more information about the ion-conduction behavior at the interface between Ni-DMF and the LPE (e.g., differential scanning calorimetry to see the shift of glass transition temperature of PVDF-HFP and/or FTIR to see the formation of C=C bonds in PVDF after Ni-DMF addition) and need to specify the multiple conduction pathways in LPE@Ni-DMF (i.e., through the LPE matrix, through Ni-DMF, and through the LPE|Ni-DMF interface). In this context, it would be valuable if the authors could measure the ionic conductivity of a Ni-DMF thin layer and/or Ni-DMF with LiFSI as this can provide useful information about which ion-conduction pathway is the most conductive and if there is a synergistic effect of mixing Ni-DMF and the LPE on conductivity enhancement.

The authors are required to address the aforementioned concerns and revise the manuscript to tell a clear story about the significance of their results in using Hofmann frameworks for solid-state batteries.

Although revising the story is the top priority, the following point-by-point suggestions would help improve the manuscript as well:

[b. Major comments]

(b-1) Lines 38–39 and 530–531: Please add the time for half cycles (either “1 h” or “0.1 mAh cm⁻²”). This is because the cycle performance depends on how much Li is deposited/stripped per half cycle, which is very important information.

(b-2) Lines 79–81: Please revise this sentence because, as I stated in (a-1-2), this appears not to be true.

(b-3) Lines 222–224: Although this sentence would be true, when we see the trajectory of the focused Li⁺ ion in Fig. 2a, it just vertically moved close to the Ni-DMF surface, which means this Li⁺ ion did not contribute to the Li⁺-ion-conduction through LPE@Ni-DME (from left to right or vice versa in the figure). It would be nice if the authors could deliver the same conclusion using a different Li⁺-DMF pair.

(b-4) Lines 232–233 and 749–751: Did the molecular dynamics simulation contain PVDF-HFP? Figs. 2a–2c and Supplementary Figs. 12 and 13 do not clearly show the contents

related to PVDF-HFP. If it has been removed from the simulation, it would mean the authors' computational result might not provide a reasonable deduction of the ion-conduction mechanisms in the real LPE and LPE@Ni-DMF.

(b-5) Fig. 2h (Lines 267-268): I am not sure if this computational result is truly informative because, in the real Ni-DMF, the lattice spacing was identified to be 7.63 Å (Fig. 1a), whereas this simulation employed nearly 8 times higher lattice spacing (i.e., 60 Å). Therefore, it is unknown whether the same phenomenon happened in the real Ni-DMF or not.

(b-6) Lines 293-294: The current rate [mA/cm²], plating/stripping amount [mAh/cm²], and temperature need to be provided.

(b-7) Supplementary Figs. 21 and 22 (Lines 300-301): How were these experiments performed? It seems no information can be found in the manuscript.

(b-8) Supplementary Fig. 26 (Lines 323-326): How can the authors ensure this sentence, although the diffraction peaks of PVDF-HFP appear at around the same 2θ values as Ni-DMF? Is there no possibility of overlapping the diffraction peaks of both PVDF-HFP and Ni-DMF?

(b-9) Supplementary Fig. 29: Why the high-frequency intercepts of both Nyquist plots are the same as each other? If LPE@Ni-DMF is more ion-conductive than the LPE, its Nyquist plots should be shifted to the left. Is there a difference in thickness between LPE@Ni-DMF and the LPE?

(b-10) Lines 378-379: The phrase "with a concentrated distribution of C, N, O, F, and S elements" needs to be revised because LPE@Ni-DMF also shows the concentrated distribution of these elements in particular regions in Supplementary Fig. 33.

(b-11) Supplementary Fig. 33: It seems the area observed for the TEM image in (a) is different from that used for element mappings in (b). Please check it and update the figure.

(b-12) Line 421: Was Fig. 4c measured by CV? If so, please specify the experimental condition carefully (e.g., voltage range, starting voltage, rest time, and if it is the two-electrode measurement, how did the authors define 0 V vs. Li/Li+).

(b-13) Lines 440-442: The theoretical capacity of SPAN needs to be specified somewhere around here. In addition, please make sure to explain the definition of 1C.

(b-14) Lines 447-451: The authors need to provide the charge-discharge curves at 150th-200th cycles (e.g., every 10 cycles) in the Supporting Information and to explain why they thought the severe capacity drop was caused by micro-short circuiting by pointing out a characteristic voltage-current response stemming from it. Additionally, dQ/dV analysis would help identify some differences between the voltage-current responses easily.

(b-15) Lines 450-451: This sentence needs a revision because the capacity fading behaviors in Fig. 5b are not the same as those in Fig. 5a. The obvious fact is that LPE@Ni-DMF showed a better capacity retention than the LPE at both 0.2C and 1C. It would be nice if the authors could revise the sentence to improve readability.

(b-16) Fig. 5d (Lines 481-483): For a fair comparison, the top half of the y-axis in Fig. 5d needs to be (retained) areal capacity [mAh/cm²], which can include the difference in the sulfur loading between the studies. If the authors can, making 3D plots of this dataset using the x-axis of (retained) energy density [Wh/kg-cathode active material], the y-axis of (retained) power density [W/kg-cathode active material], and the z-axis of the maximum cycle number would identify the best performance more clearly.

(b-17) Lines 558-559: This procedure needs to be explained more clearly. For instance, what were the weights of a DMF solvent and Ni-activated?

(b-18) Materials preparation: The supplier and grade information of each material should be provided. These materials are:

- NiCl₂·6H₂O
- K₂[Ni(CN)₄]
- DMF
- PVDF-HFP
- LiFSI
- SPAN
- Multi-walled carbon nanotubes
- Al foil (please include the information about its thickness)
- Li metal
- Al tab
- Ni tab
- Al compound packing film

(b-19) Lines 562-568: This procedure needs to be explained more clearly. Please provide the following information:

- The gap between the doctor blade and the glass substrate
- For LPE@Ni-DMF:
 - o The weight ratios between Ni-DMF, PVDF-HFP, LiFSI, and DMF before casting
 - o The order of addition (e.g., weighed PVDF-HFP first, then LiFSI was added onto it followed by the addition of DMF. After stirring the mixture at 80 °C for 4 h, Ni-DMF was added and another 1 h mixing was performed)
 - o Mixing time after Ni-DMF addition

(b-20) Lines 597-599: As I stated in (a-3), please include the LSV data of each material in the manuscript.

(b-21) Lines 614-617: The authors need to provide more information about the pouch-cell assembly. For instance:

- The size of each electrode
- The size of an LPE@Ni-DMF sheet
- The size of each tab
- The type of the tab used for each electrode (I presume an Al tab for SPAN/Al and a Ni tab for Li metal)
- How did the author attach a tab onto each electrode (used a tab welding machine?), especially for the Li metal anode?

(b-22) Line 637: The authors need to explain how the Li metal tablet was washed in detail.

(b-23) Line 727: This current density is exceptionally high compared to these applied in the

actual experiments (100 vs. 0.1–1 mA cm⁻²). Is there any justification?

(b–24) Supplementary Table 1: The reference for the first row might be 13 and that for PVBL might be 15. In the caption, the name of the table needs to be “Supplementary Table 1” (please remove “S” before “1”). In the footnotes, please explain all abbreviations (e.g., PMLSE, HSE-etched bm-LLZO30, PHLC (20% CAP), etc.).

[c. Minor comments]

(c–1) Lines 70–73: To make sure of the authors’ claim, it would be nice if the authors could specify the concentration (or weight ratio) of the diluted DMF molecules in Ref. 10 vs. that of another study without using any additive.

(c–2) Lines 111 and 352: The word “overpotential” would need to be revised as “overvoltage” because this is a two-electrode measurement and the voltage reading is the difference between the electrode potential between the two Li-metal electrodes.

(c–3) Lines 119–120: It would be helpful to specify the areal capacity in mAh/cm² as well.

(c–4) Line 143: The article “The” would be changed to “This” for better readability.

(c–5) Line 157: The phrase “Stack date” would be “Stack data”.

(c–6) Line 200: In the caption for Fig. 1e, “L-edge” would be “L3-edge”.

(c–7) Line 257: The word “aggregate” would be changed to “concentrate” because “aggregate” would make readers imagine the aggregation (to form big particles).

(c–8) Line 345: The phrase “active anode materials” would be changed to “the anode active material”.

(c–9) Line 455: I think “up to” is redundant.

(c–10) Line 474: I think it is better to rephrase “LPE@Ni-DMF-SEI” to a clearer explanation (e.g., simply, “the SEI layer” because readers can understand this is the case for LPE@Ni-DMF by reading the clause before “that”).

(c–11) Lines 488–492: It would be nice if the authors could also specify the capacity retention of the SPAN/LPE/Li cell in percentage.

(c–12) Line 541: Maybe the authors would want to add “SPEs for” between “...extended to” and “other emerging solid-state battery systems” because I believe the authors’ approach is for the development of SPEs and is not directly for the emerging solid-state batteries (e.g., electrodes, cell-enclosure, and operational condition designs, etc., which are outside the scope of this study).

(c–13) Line 584: Please specify the direction of the experiment (e.g., tested from the highest frequency to the lowest frequency).

(c–14) Line 586: The symbol “Ea” should be “E_a” (“a” should be subscripted). Please check the other parts as well.

(c-15) Line 589: The phrase "Li/SSEs/Li cells" would need to be "Li/SSE/Li cells". Please check the other parts as well.

(c-16) Line 590: Please cite the reference that proposed this equation.

(c-17) Line 593: The symbol "Rs" should be " R_s " ("s" should be subscripted).

(c-18) Lines 609-610: I think the authors would need to specify the test temperatures as they studied multiple temperatures.

(c-19) Lines 624-625: The authors would want to add "(the) thermal stability" here because this is the data that TG analysis can provide but it has not been specified.

(c-20) Line 697: The symbol " Φ_e " would need to be " Φ_I ".

Point-by-Point Response to the Reviewers' Comments

REVIEWER COMMENTS

Reviewer #1 (Remarks to the Author):

This research aimed to address the issues related to the low ionic conductivity of solid polymer electrolytes and the instability of lithium-metal in DMF with high polarity. It tackled these challenges by utilizing the ligand-assisted transport mechanism and conducted extensive simulations to provide insights into these solutions. However, there are some unconvincing parts in terms of interpreting the experimental results. It seems that an ambiguous explanation is being made to connect the experimental results. Several inherent concerns were unavoidable before supporting this work for publication.

1. Looking at the simulation data, it seems that it takes a relatively long time for DMF to be included in the Li ion first solvation structure (0.76ns) and then return to the Ni-DMF crystal structure (36.79ns). If DMF does not bind well for a long period of time, it feels like there is enough time for DMF diffusion to occur. What do you think about this?

Response:

We thank the reviewer for bringing up this important point. We agree that the time and distance of the simultaneous migration of Li⁺ and DMF is essential for determining the rate of DMF shuttling. It is noteworthy that the detrimental effect stemming from DMF diffusion occurs predominantly during electrochemical cycling rather than calendar aging. Hence, we underscore that the shuttling of [DMF-Li⁺] clusters during electrochemical cycling is more harmful to Li metal anode compared to the diffusion of free DMF molecules. Hence, we primarily focus on resolving the correlated transport behavior of DMF and Li⁺. Recognizing the limitations of the originally presented single snapshot, which may not comprehensively capture the dynamics of the overall bulk average, we have performed further calculations and statistical analyses on this aspect.

The diffusion of Li⁺ concerning another species can be characterized into two primary modes: vehicular and structural. In the former, Li⁺ diffuses alongside its solvation shell as a single complex, whereas in the latter, neighboring species do not move together for appreciable distances (*ACS Cent. Sci.* **5**, 1250-1260 (2019)), leading to frequent exchanges within the Li⁺ solvation shell. The diffusion mechanism of one species relative to another can be distinguished quantitatively by calculating the residence time (τ) for two neighboring species to move together. In this context, we have evaluated residence times for Li⁺ with respect to DMF and FSI⁻ (**Supplementary Fig. 15**). We observe a notable reduction of 60% in the residence time for both Li⁺/DMF and Li⁺/FSI⁻ pairs within LPE@Ni-DMF compared to pristine LPE. Furthermore, we can use this quantity to convert from residence time to the diffusion length of DMF co-migrating with Li⁺, calculated as $L = \sqrt{6D_{DMF}\tau}$. As shown in **Supplementary Fig. 17**, the diffusion coefficients for DMF (D_{DMF}) and FSI⁻ (D_{FSI^-}) remain relatively constant for both LPE and LPE@Ni-DMF. Therefore, both the residence time (τ) and the diffusion length (L) of DMF moving together with Li⁺ are significantly reduced, which further elucidates the mitigating effect on DMF shuttling within LPE@Ni-DMF.

Accordingly, the following discussion has been added to the revised manuscript (**Page 12**):

“We observe a notable reduction of 60% in the residence time for both Li^+ /DMF and Li^+ /FSI⁻ pairs within LPE@Ni-DMF compared to pristine LPE, in accordance with the lower activation energy (E_a) of LPE@Ni-DMF (0.147 eV) than that of LPE (0.218 eV; Supplementary Fig. 15).” and “The diffusion length of DMF co-migrating with Li^+ can be determined utilizing the equation $L = \sqrt{6D_{\text{DMF}}\tau}$. Since the diffusion coefficients for DMF (D_{DMF}) and FSI⁻ (D_{FSI^-}) remain relatively constant in LPE and LPE@Ni-DMF (Supplementary Fig. 17), both the residence time and diffusion length of DMF moving concurrently with Li^+ exhibit significant decrease within LPE@Ni-DMF, which mitigates the DMF shuttling.”

Supplementary Fig. 16 | Arrhenius plots of LPE and LPE@Ni-DMF at different temperatures.

Supplementary Fig. 17 | The diffusion behavior of Li^+ , DMF, and FSI⁻ in LPE and LPE@Ni@DMF. (a) Mean square displacements (MSD), and (b) self-diffusion coefficients (D).

2-1. You mentioned the incomplete reduction product of SO_2F in the SEI component. What do you think is the difference between your electrolyte and the control group? Because FSI is strongly captured and fixed by Ni^{2+} , it is thought that it will be difficult for anions to reach the electrode surface.

Response:

Thank you for your insightful questions about the role of SEI in our work. The FSI⁻ can be degraded in both cases of LPE and LPE@Ni-DMF due to their directly contacting with the Li metal interface. However, the key difference between the LPE@Ni-DMF and the control sample is the ability of Ni-DMF to effectively immobilize DMF. In the control sample, DMF is not immobilized, which tends

to degrade on the Li metal anode and leads to the formation of a SEI layer with an uneven distribution of organic components. The excessive presence of organic phase in the SEI hinders subsequent Li^+ transport, making the anode more susceptible to dendrite growth. In contrast, the LPE@Ni-DMF yields a higher content of inorganic compounds with higher conductivity within the SEI layer due to the inhibition of DMF decomposition at the Li metal anode interface. Meanwhile, the Li anode is also protected from corrosion by DMF, resulting in a more stable electrode-electrolyte interface.

We also observe the immobilization of FSI^- in LPE@Ni-DMF (**Supplementary Fig. 14**), which together with enhanced structural diffusion of Li^+ contribute to the enhanced transference number. However, the depletion of anions on the anode interface is unlikely to occur. Firstly, the calculated adsorption free energy of the FSI^- anion on $[\text{Ni}(\text{CN})_4]^{2-}$ site is smaller than that of DMF adsorption (**Supplementary Fig. 14 and 29**), indicating a stronger confinement effect of DMF over FSI^- . Secondly, the residual DMF in LPE@Ni-DMF accounts for 12.2 wt % (**Fig. 3f**), whereas the weight percentage of LiFSI is approximately 43 wt % (LiFSI:PVDF-HFP = 1:1). Thus, the mass and molar ratio of FSI^- in the bulk electrolyte are subsequently greater than those of DMF. Finally, according to Debye–Hückel theory, the electrolyte system maintains an average electrical neutrality. Hence, any local deficiency of FSI^- can only persist for a limited distance, typically on the order of several Å. We hope that our answer can well resolve your concern.

2-2. In general, in liquid electrolytes, a high transference number indicates that more anions are included in the lithium-ion first solvation structure. However, your paper states the opposite. What do you think about the relationship between the lithium hopping mechanism and inorganic-based SEI components?

Response:

Thank you for carefully reviewing our manuscript and for highlighting the important issue regarding the interpretation of the Li^+ transference number.

Indeed, an increasing number of anions can enhance Li^+ migration kinetics in liquid electrolytes by weakening the lithium-solvent interactions and promoting lithium hopping, especially in high concentration electrolytes (*ACS Energy Lett.* **9**, 373–380 (2024)). However, our strategy involves weakening Li^+ coordination strength by incorporating Hofmann frameworks. The interactions between DMF and open metal sites of the Hofmann framework weaken the local lithium-solvent interactions in LPE@Ni-DMF compared to pristine LPE. This is evidenced by an increased Li-O(DMF) distance (**Fig. 2e and 2f**), an increased Li^+ self-diffusion coefficient (**Supplementary Fig. 17**), and decreased residence times for Li^+ with respect to DMF and FSI^- (**Supplementary Fig. 15**), thereby providing rapid Li^+ transport pathways.

Regarding the relationship between the lithium hopping mechanism and inorganic-based SEI components, we believe that promoted lithium hopping in the bulk electrolyte is beneficial for the formation of an inorganic-rich SEI. As elaborated in the response to question 1, Li^+ diffusion can be characterized into vehicular and structural (hopping) modes. With a strong lithium-solvent interaction, the vehicular mode prevails, resulting in the continuous shuttling and reactions of $[\text{DMF-Li}^+]$ clusters. In contrast, with a weakened lithium-solvent interaction, the structural (hopping) diffusion plays a major role, leading to a quick depletion of $[\text{DMF-Li}^+]$ at the anode surface and less organic reaction products in the SEI. We hope that these explanations address your concern satisfactorily.

2-3. Could you provide the pick intensity of XPS data?

Response:

Thanks for the reviewer's comment and the pick intensity of XPS data have been updated in **Supplementary Fig. 36 and 37.**

Supplementary Fig. 36 | X-ray photoelectron spectroscopy O1s depth profiles (by sputtering for different times) of cycled Li metal anode using LPE and LPE@Ni-DMF and corresponding atomic ratio change.

Supplementary Fig. 37 | X-ray photoelectron spectroscopy F1s depth profiles (by sputtering for different times) of cycled Li metal anode using LPE and LPE@Ni-DMF and corresponding atomic ratio change.

2-4. And how can the charge imbalance between the two electrodes resulting from immobilization of anions be interpreted?

Response:

Thank you for raising the question concerning the charge conservation rule in batteries due to the immobilization of anions.

It is important to note that anion immobilization does not inherently conflict the principle of charge conservation in a battery electrolyte.

During the transport of Li^+ from one electrode to another, a net negative charge remains on the source electrode (anode during discharge, cathode during charge), while Li^+ accumulates at the destination electrode (cathode during discharge, anode during charge). However, this process does not necessarily generate a potential difference across the battery. Concurrently, there is a net current flow through the external circuit that transports counter charges to the destination electrode, balancing with Li^+ . In fact, the concept of single-ion conductor is a prominent area of focus in solid-state electrolyte research (*Nat. Commun.* **9**, 5029 (2018); *J. Energy Chem.* **81**, 313 (2023)), and anion immobilization is a commonly utilized strategy to achieve a high transference number (ideally 1). We hope that our answer can well resolve your question.

3. It was said that DMF reactivity and low ionic conductivity were solved simultaneously, but most

of the data was interpreted based on simulation. Can the above results (Especially, Reduction intensity of solvent decomposition peak in CV.) be obtained through electrochemical experiment?

Response:

Thanks for the reviewer's comment. We indeed conducted extensive simulations to gain insights into ion/or cluster transport behavior in LPE and LPE@Ni-DMF. The observed ligand-assisted transport mechanism improved the Li⁺ conduction with the utilization of Ni-DMF. This was confirmed by the ionic conductivity test, where LPE@Ni-DMF demonstrated a higher ionic conductivity of $6.5 \times 10^{-4} \text{ S cm}^{-1}$ at room temperature than that of LPE ($2.4 \times 10^{-4} \text{ S cm}^{-1}$).

The experimental verification of the weakened of [DMF-Li⁺] cluster shuttling was confirmed by TG analysis, where the relative weight change after cycling reduced from 7.5 wt % for LPE (**Fig. 3h**) to 1.7 wt % for LPE@Ni-DMF (**Fig. 3i**). These results visually confirmed the reduced DMF consumption in LPE@Ni-DMF during electrochemical reaction.

The Li//Li symmetrical cell test has also demonstrated the significant role of Ni-DMF in reducing DMF shuttling. The Li/LPE/Li cell displayed increasing overvoltage and interfacial resistance after only 500 hours of cycling, which is due to the deteriorative SEI layer caused by DMF corrosion. In contrast, the Li/LPE@Ni-DMF/Li cell stably cycled for 6,250 hours with negligible increase in overvoltage. The electrochemical results indicate that DMF reactivity was reduced and that DMF was strongly confined in LPE@Ni-DMF.

Additionally, the DRT analysis of in-situ EIS measurements confirms a similar conclusion. The increase and the rightward shift in impedance assigned to SEI at 10^{-4} s in LPE can be attributed to the continuous growth of the organic-rich SEI layer, which is associated with the continuous reaction between DMF and Li metal. However, the impedance in LPE@Ni-DMF remained nearly unchanged, demonstrating that the SEI layer with inorganic compounds as dominant contents is more stable once formed. The electrochemical results also account for the fact that the Ni-DMF additives induce a stable SEI layer, which prevents the continuous reaction between DMF and Li metal.

The cyclic voltammetry (CV) curves of SPAN/SSE/Li cells in the first three cycles were obtained using a scanning rate of 0.1 mV s^{-1} at $1.0 \sim 3.0 \text{ V}$ (**Fig. R1**). No obvious decomposition peak of DMF was found in either SPAN/LPE/Li or SPAN/LPE@Ni-DMF/Li cells. This phenomenon can be explained that because of the strong interaction between LiFSI and DMF, a short-term stable interface with Li metal could be obtained, but prolonged cycling or higher current densities lead to a gradual increase in the reaction with Li metal.

In summary, all the above experimental results demonstrated that the DMF reactivity and low ionic conductivity issues were simultaneously solved by Ni-DMF additives. We hope that our answer can well address your concerns.

Fig. R1 | The CV curves of SPAN/LPE/Li (a) and SPAN/LPE@Ni-DMF/Li (b) cells.

Reviewer #2 (Remarks to the Author):

This is a beautiful mechanistic study of how Ni bound DMF can increase the ionic conductivity of polymer electrolytes and at the same time not to be consumed by reaction with Li anode. The study has many complimentary tools such as TGA, SAXS/WAXS, TEM and simulation to substantiate their conclusions. This work can be published as is. I just have a minor question - how do the authors determine how much Ni-DMF to incorporate into the LPE? Does the starting LPE and CPE have similar DMF content?

Response:

We appreciate the reviewer's positive feedback and support for our work being published in Nature Communications.

Actually, the additive amount of Ni-DMF in LPE@Ni-DMF preparation is 60 mg and the weight ratios between Ni-DMF, PVDF-HFP, LiFSI, and DMF is measured to be 1:5:5:47. We apologize for the missing information, which has been added in the revised manuscript ("Materials preparation" section, **Page 31**) as described below:

"With other conditions unchanged, 60 mg of Ni-DMF powder was added to the mixture of PVDF-HFP and LiFSI solution with another 2 hours mixing performed before the casting process (the weight ratios between Ni-DMF, PVDF-HFP, LiFSI, and DMF were measured to be 1:5:5:47), and the LPE@Ni-DMF membrane can be obtained."

We also prepared control samples with 30 mg and 90 mg of Ni-DMF additives, denominated as CPE30 and CPE90, respectively. CPE30 exhibited higher ionic conductivity of $3.4 \times 10^{-4} \text{ S cm}^{-1}$ than that of LPE ($2.4 \times 10^{-4} \text{ S cm}^{-1}$), indicating that the ligand-assisted Li^+ transport mechanism at the Ni-DMF–electrolyte interfaces is beneficial for improving ionic conductivity (**Fig. R2**). In contrast, the CPE90 exhibited the lowest ionic conductivity of $1.6 \times 10^{-4} \text{ S cm}^{-1}$ among the experimental and control samples, which could be caused by aggregation of Ni-DMF additive. The SEM image of CPE90 can further support our statement where the Ni-DMF nanosheets exhibit an uneven distribution in complex matrix (**Fig. R3**). The Ni-DMF aggregation is not only harmful for the spatial uniformity of ion transport in the electrolyte but also affects the distribution of electric field when the electrolyte contacts a Li metal anode. Therefore, we determined that 60 mg represents the optimal Ni-DMF content in the blended polymer electrolyte.

The DMF content in fresh LPE is 13.4 wt %, which is similar to that of fresh CPE (12.2 wt %) (**Fig. 3h and 3i**).

Fig. R2 | Nyquist plots of CPE30 and CPE90 at room temperature.

Fig. R3 | SEM image of CPE90. The region in the white dotted square shows uneven distribution of Ni-DMF nanosheets.

Reviewer #3 (Remarks to the Author):

Zhu et al. created a Hofmann-DMF coordination complex and incorporated it into a PVDF-HFP matrix to obtain a composite-polymer-electrolyte (CPE) membrane labeled as LPE@Ni-DMF for solid-state batteries. The LPE@Ni-DMF showed the ionic conductivity of $6.5 \times 10^{-4} \text{ S cm}^{-1}$ at room temperature, which is an acceptable level given the classification of solid electrolytes (i.e., CPEs), and enabled stable cycling of the Li|Li symmetric cell at 0.1 mA cm^{-2} , 0.1 mAh cm^{-2} ($\approx 2 \text{ h/cycle}$) for $\geq 3,000$ cycles. The authors deduced the origins of the long cycling stability of LPE@Ni-DMF are improved Li^+ conductivity and uniformly distributed Li^+ ions at the Li|solid-electrolyte interface, which allowed the smooth deposition of Li metal. Their experimental and computational data support this consideration. They also demonstrated the applicability of this solid-electrolyte membrane to solid-state lithium-sulfur batteries using a sulfurized polyacrylonitrile (SPAN) cathode with a sulfur loading of 1 or 2 mg cm^{-2} in both coin-cell and pouch-cell formats, of which the coin

cell maintained a specific capacity of ≥ 500 mAh/g (i.e., ≥ 0.5 mAh/cm²) for $\geq 1,000$ cycles (at room temperature, 1C).

Overall, the results look great; Ample experiments using various techniques have been conducted from material characterization to performance evaluation and its reasoning, which will definitely be informative for researchers in the battery field.

However, at this stage, I hesitate to provide a positive recommendation for this manuscript given (a-1) the hitherto progress of the research field, (a-2) the mismatch between the focused issue in the manuscript and the authors' approach, (a-3) unknown electrochemical stability of the authors' Hofmann frameworks, (a-4) undisclosed issue associated with Ni species coming from the authors' Hofmann frameworks, and (a-5) unspecified effect of Ni-DMF addition on the interfacial properties of the LPE. These concerns are shown in detail as follows:

[a. Major Concerns]

(a-1) Is the residual DMF in a polymer matrix really a serious problem?

In the Introduction section, the authors point out the focused issue in the manuscript: DMF decomposition on Li by saying "The uncontrolled decomposition and continuous depletion of DMF are highly detrimental to the stability of the Li metal anode as well as the durability of SPEs." (Lines 62-65) After that, they try to signify the importance of their work by stating: "In brief, previous additive engineering approaches mainly focus on tackling one issue of either ionic conductivity or electrochemical stability. An effective strategy to simultaneously improve both aspects needs to be developed for the ultimate application of SPEs." (Lines 81-85)

However, based on the provided references and information from other papers, I could not justify the decomposition of DMF trapped in a polymer matrix as a major obstacle to the solid-state-battery application of SPEs.

Response:

Many thanks for your comments. The presence of residual DMF in a polymer matrix, especially in solid polymer electrolytes (SPEs) used in solid-state batteries, is indeed a significant issue that affects both battery stability and performance (*Adv. Energy Mater.* **12**, 2200967 (2022); *J. Am. Chem. Soc.* **145**, 25632-25642 (2023); *Nano Lett.* **18**, 6113-6120 (2018)). Uncontrolled decomposition and continuous depletion of DMF can harm the stability of the Li metal anode and the durability of SPEs. We further elaborate the negative effects of the residual DMF by the following points:

(1) Electrolyte-Electrode Interface Stability

DMF is known to be chemically reactive towards Li metal. This reaction can cause the formation of an unstable SEI when DMF comes into contact with the Li metal anode. The instability can further lead to continuous consumption and depletion of DMF, resulting in poor cycling performance and reduced battery lifespan. In this work, we employed COMSOL simulations to model the concentration evolution of DMF. The simulations demonstrated that under an applied external electric field, DMF migrates towards the Li metal anode, where it reacts and decomposes. This resulted in a continuous decrease in DMF concentration in the SPE (**Fig. 4g-j**). The decomposition of DMF can lead to the formation of unsatisfactory byproducts at the electrode-electrolyte interface. The byproducts constructing SEI are mainly organic composites which form resistive layers,

impeding ion transport and contributing to increased internal resistance within the battery (**Fig. 4d**). This can result in capacity fade and reduced efficiency of battery over time.

(2) Durability of SPEs

DMF is frequently utilized as a plasticizer in SPE to improve ionic conductivity by aiding in the dissociation of lithium salts. The degradation of DMF not only impacts the stability of the interface but also reduces the amount of effective plasticizer in the SPE matrix. The change in relative DMF content after cycling decreased from 7.5 wt % for LPE (**Fig. 3h**) to 1.7 wt % for LPE@Ni-DMF (**Fig. 3i**). Over time, the ionic conductivity of LPE may decrease, which can negatively impact the battery's performance.

In summary, the degradation of DMF within the SPE matrix is a critical issue that hinders the practical application of SPEs in solid-state batteries. It affects the stability of the electrolyte-electrode interface, leads to the depletion of plasticizer thereby reducing ionic conductivity, and results in the formation of detrimental byproducts. The significance of this work lies in proposing a novel approach that addresses the challenges by introducing a design that confines DMF and improves both ionic conductivity and electrochemical stability.

(a-1-1) For instance, in Ref. 9, by which the authors claim that DMF decomposition on Li is serious (Lines 60–62), the opposite opinion can be found, where the authors of Ref. 9 say: “The strong [DMF-Li⁺] complex in the PVDF-DMF-LiFSI film eliminates the free DMF that can attack the metallic lithium. Moreover, it was reported that the stable solid electrolyte interphase (SEI) layer between the PVDF-DMF-LiFSI film and the lithium metal can suppress the lithium dendrite growth” with the 38 quoted references therein. Indeed, they show the stable Nyquist plot of the Li|PVDF-DMF-LiFSI|Li symmetric cell for at least 216 h and the stable cycling of a solid-state lithium-metal battery (i.e., Li|PVDF-DMF-LiFSI|LiNi_{1/3}Co_{1/3}Mo_{1/3}O₂) with no sign of capacity decay for at least 200 cycles at room temperature, 0.2C. Therefore, I am reluctant to say DMF decomposition needs to be addressed urgently.

Response:

We appreciate the reviewer's thoughtful comment. The observations in Ref. 9 regarding [DMF-Li⁺] complex improving the stability of the PVDF-DMF-LiFSI system compared to free DMF in solution for liquid-state electrolyte systems. However, this does not necessarily contradict the fact that DMF decomposition is still a significant concern in SPE, as decomposition still occurs during lithium deposition due to the participation of DMF in Li⁺ coordination, especially during prolonged cycling. The experimental behavior of the battery is highly dependent on specific conditions, such as the concentration of components and the current density during operation. In particular, Nyquist plots of the Li//Li cell was tested after 216 hours of lithium plating and deposition with a static impedance improvement from 175 Ohm to 210 Ohm, which may also be due to the deterioration of Li⁺ transport phenomenon in SPE caused by DMF decomposition. Our research shows that the battery experiences sudden shutdowns and a decrease in capacity. This suggests that DMF decomposition has a significant impact on long-cycle performance. We speculate that the SPE in Ref. 9 have not yet completely depleted DMF after 200 cycles, despite having a DMF content of 15% and delivering a capacity of 0.3 mAh cm⁻². Addressing DMF decomposition may be more relevant for ultra-long-cycle systems or high areal loading conditions in which the batteries are expected to operate for future practical use. We hope that our explanations have addressed your concerns adequately.

(a-1-2) In addition, as for the CPEs containing dielectric ceramic materials, the authors state “However, due to the limited amount and surface area of additives, their ability to immobilize DMF, particularly during calendar aging, is insufficient.” (Lines 79–81) by mentioning Refs 13–15. However, these references, especially Refs 14 and 15 do demonstrate an exceptionally long cycling performance for both symmetric and asymmetric cells (=lithium-metal batteries). For the cycle stability of Li|Li symmetric cells, Ref. 14 shows $\geq 1,300$ cycles at 0.1 mA cm^{-1} , 0.1 mAh cm^{-1} (=2 h/cycle), while Ref. 15 shows ≥ 950 cycles at 0.1 mA cm^{-1} , 0.1 mAh cm^{-1} (=2 h/cycle), $25 \text{ }^\circ\text{C}$. As for the full cells, Ref. 14 shows the long-term cycling of Li|LiNi_{0.8}Co_{0.1}Mo_{0.1}O₂ with a capacity of $\geq 100 \text{ mAh/g}$ (i.e., $\geq 0.16 \text{ mAh/cm}^2$) for $\geq 1,500$ cycles at 2C, $25 \text{ }^\circ\text{C}$, while Ref. 15 shows the Li|LiNi_{0.8}Co_{0.1}Mo_{0.1}O₂ cell that can be cycled at 1C, $25 \text{ }^\circ\text{C}$ for $\geq 1,500$ cycles with a capacity of $\geq 100 \text{ mAh/g}$ (i.e., $\geq 0.16 \text{ mAh/cm}^2$). The difference in cell performance between them and the authors’ results is less obvious and, therefore, it can be deduced that a continuous DMF decomposition on the Li side would not be detrimental to the long-term stability of SPEs.

Response:

We appreciate the reviewer's careful review. Your points regarding the cycle stability of cells mentioned in Refs. 14 and 15, which exhibit exceptional long-term cycling performance, are valid and noteworthy. These references indicate that certain CPEs containing dielectric ceramic materials can exhibit excellent long-term cycling performance in both symmetric (Li//Li) and asymmetric cells (full cells with lithium-metal batteries). While Refs. 14 and 15 show impressive cycling performance, our work demonstrates stable cycling for over 6,000 hours (2 h/cycle) at 0.1 mA cm^{-2} in Li//Li symmetric cells. This highlights the need for a comprehensive solution that addresses ionic conductivity and electrochemical stability issues of SPEs performance. Regarding full cells, the performance of a battery system is not solely determined by individual components, but rather by how they function together. Previous studies have primarily reported performance metrics, such as cycles, current density, and capacity retention, based on LFP or NCM cathodes. A sulfur-based cathode has the potential to deliver a higher specific capacity, which could improve the energy density in future practical use. Therefore, we demonstrated the applicability of the solid-electrolyte membrane to solid-state lithium-sulfur batteries using a SPAN cathode with a sulfur loading of 1 mg cm^{-2} in coin-cell format. The coin cell maintained a good specific capacity of $\geq 500 \text{ mAh g}^{-1}$ (i.e., $\geq 0.5 \text{ mAh cm}^{-2}$) for $\geq 1,000$ cycles at room temperature and 1C.

However, during cycling, the sulfur-containing cathode undergoes significant volume changes and experiences sluggish kinetics, which can limit the overall performance and stability of the battery. To improve the full cell in the future, it is necessary to optimize the mass production parameters or design the cathode's tortuosity/porosity and protective Li metal.

Accordingly, the statement of “However, due to the limited amount and surface area of additives, their ability to immobilize DMF, particularly during calendar aging, is insufficient.” was revised to “However, previous approaches to additive engineering have mainly focused on addressing either ionic conductivity or electrochemical stability. The synergetic effect of DMF modulation and Li⁺ transport mechanism among the complex components remains unclear, which substantially impedes the design of high-performance and durable polymer electrolytes. In brief, an effective strategy to simultaneously improve both aspects needs to be developed for the ultimate application of SPEs.”

(Page 4)

(a-1-3) Furthermore, a vast number of past studies on the CPEs using metal-organic frameworks,

to which the authors' current work might belong, have already exhibited excellent ionic conductivities of such CPEs and competitive symmetric and asymmetric cell performances, discoloring the significance of the authors' results to some extent. Please take a close look at the following review papers in this field and references therein (please cite them if relevant to do so in the revision stage):

- Interdisciplinary Materials 2 (2023) 475–510
- Electrochemistry Communications 150 (2023) 107491
- EcoMat 5 (2023) e12283

Response:

Many thanks for your suggestion. As you mentioned, metal-organic frameworks (MOFs) have been widely used as additives in the field of CPEs in recent years, with excellent ionic conductivities and cell performances reported. However, most MOFs-based CPEs utilize the structural advantage of MOFs, where the inner macropores serve as a 2D channel for ionic clusters transport. Some recent efforts involve utilizing crafted open metal sites within MOFs to anchor anions as mentioned by the reviewer. In our work, we emphasized the significance and the underlying mechanism of the residual solvent for Li salt dissociation, which leads to higher ionic conductivity. However, we also paid attention to the adverse impact of the solvent on Li metal. Therefore, we proposed an integrated design that combines Hofmann frameworks and DMF (namely Hofmann-DMF) to regulate the coordinated environment of Li⁺ for fast ion transport and to mitigate the residual solvent shuttling for a stable Li metal anode.

We carefully read the references mentioned and cited the review papers "*EcoMat* 5, e12283 (2023)" as Ref. 10, "*Interdisciplinary Mater.* 2, 475–510 (2023)" as Ref. 21, and "*Electrochem. Commun.* 150, 107491 (2023)" as Ref. 22, respectively, in the revised manuscript (Page 4 and Page 5).

(a-1-4) It seems the authors' concept (i.e., the use of open metal sites of a framework to trap solvent molecules, thereby mitigating the issue associated with the solvent decomposition on Li) has been already proven by this past study: *Journal of Power Sources* 240 (2013) 653–658, where Zn₄O(BDC)₃ (BDC stands for 1,4-benzenedicarboxylate) was incorporated in a poly(ethylene oxide) (PEO) matrix with residual DMF molecules, and the main cause of the improved reversibility of the Li plating/stripping behavior and the improved rate and cycle performances of Li|LiFePO₄ solid-state cells were concluded as the result of the strong solvent-trapping ability of Zn₄O(BDC)₃. Although the used materials in this past study are different from the authors' work, it has demonstrated the same concept. Therefore, the authors would need to provide other evidence to clarify the originality of the work.

Response:

Many thanks for your suggestion. Metal-organic frameworks have been demonstrated mitigate reversibility issues, as evidenced by the incorporation of Zn₄O(BDC)₃ into a PEO matrix in the *J. Power Sources* article. However, the mechanism behind interface stability and lithium transportation may still be unclear. For example, in the mentioned paper, the authors mainly attribute the "Lewis-acidic sites on MOF-5" to the interaction "with PEO chains and lithium salt". Any further characterizations (e.g., surface XPS) and mechanistic justifications on the "effect that trace solvent absorbed in porous filler" were not provided. Although it is clear that the MOFs additives have proven ability to accommodate solvent molecules, it is worth exploring further on designing well-defined solvent confining structures with verifiable solvent shuttling prevention effects, especially

during prolonged cycling for practical application. Therefore, providing detailed insights into the mechanism of solvent trapping and its impact on the battery's performance, particularly clarify the synergistic influence of DMF decomposing on anode stability and durability of SPEs can be a significant contribution. In this study, we use a combination of experimental and FEM simulation methods to investigate the dynamic evolution process of SEI and the corresponding Li⁺ deposition behavior under cycling conditions, and we systematically investigate the interaction and the resultant effects of incorporating Hofmann-DMF in lithium coordination environment on battery behavior.

Other than the mentioned paper, some recent studies have reported various additives to promote salt dissociation and immobilize DMF (as referred in the main text). However, their ability to tune the lithium coordination environment by modifying the DMF state is insufficient, particularly during calendar aging or at low temperature. The influence on lithium coordination and transport among the multi-phase interface of SPEs needs to be further elucidated. The Hofmann-DMF complex exhibits a unique 2D characteristic that can provide ample interface for rapid diffusion of Li⁺, thereby enhancing Li⁺ transport efficiency. More importantly, by engineering DMF into the Hofmann coordination complex, surface-tethered DMF ligands can not only contribute to the solvation structure of Li⁺ and but also promote Li⁺ transport by creating a locally solvent-rich environment. The environment rich in solvent weakens the coordination strength of lithium, resulting in an increased Li-O(DMF) distance (**Fig. 2e and 2f**). This demonstrates a weaker interaction between Li⁺ and DMF in LPE@Ni-DMF compared to pristine LPE. Furthermore, the calculated residence time and diffusion length of DMF moving concurrently with Li⁺ exhibit significant decrease within LPE@Ni-DMF (**Supplementary Fig. 15 and 17**), which contribute to the mitigated DMF shuttling. Further details on the calculation process are included in the response to question a-2-1.

(a-2) Does mixing Ni-DMF with the LPE really help mitigate the decomposition of DMF molecules that are in the LPE matrix?

Response:

We appreciate the reviewer's thoughtful comment. We believe that the Ni-DMF can effectively mitigate the decomposition of DMF molecules involved in the LPE matrix. The detail explanations are included in the following responses.

(a-2-1) Based on Figs. 3a and 3b, and Supplementary Videos S1 and S2, readers can identify that LPE@Ni-DMF has Ni-DMF grains that are fully covered with the LPE and the LPE forms 3D ion-conduction network throughout LPE@Ni-DMF. This means that the incorporation of Ni-DMF into the LPE cannot prevent the LPE from being exposed to the Li metal side. Therefore, regardless of Ni-DMF incorporation, the LPE that contains residual DMF molecules in its matrix definitely attaches to Li metal. In this context, the authors' results cannot be direct proof of the mitigated decomposition of DMF molecules in the LPE matrix using Ni-DMF.

Therefore, the authors' approach appears to be doing the same thing as additive engineering where the effect of DMF decomposition on cycle stability was minimized because the concentration of trapped DMF in a polymer matrix per unit volume was diluted. This dilution effect is also valid in the authors' case; Even though DMF molecules in Ni-DMF are immobilized greatly, those trapped

in the PVDF-HFP matrix have no bearing on immobilization. Hence, the stable cycling of Li|Li symmetric cell using LPE@Ni-DMF appears to benefit from the diluted amount of trapped DMF molecules in a polymer matrix in a unit volume, and, therefore, the authors' approach does not like the significant breakthrough to mitigate the adverse effect stemming from the decomposition of DMF molecules trapped in a polymer matrix.

Response:

We appreciate the reviewer's thoughtful feedback. The primary role of the Ni-DMF complex in the LPE matrix is not to function as a physical barrier between the LPE and Li metal, but rather to stabilize DMF molecules through well-defined microscopic interactions. The Ni-DMF complex functions to anchor DMF molecules, reducing their mobility and the probability of their migration to the Li metal surface where they would decompose. Although it is not possible to completely prevent contact between DMF and the Li metal, coordinating DMF as Hofmann ligands could alter its coordination environment, making it less reactive towards anode and thereby mitigating its continuous decomposition compared to the shuttle effect of [DMF-Li⁺] in pristine LPE. This conclusion is supported by the larger Li-O(DMF) distance (2.11 Å) in LPE@Ni-DMF compared to that in LPE (2.05 Å; **Fig. 2e and 2f**). The DMF ligand of Ni-DMF involved in the solvation structure of the Li⁺ in LPE leads to an increase in the number of solvent molecules surrounding Li⁺ and, consequently, a longer Li-O bond length. This weakens the interaction between Li⁺ and DMF molecules, leading to promoted Li⁺ transport and the easier desolvation of Li⁺ at the anode interface. This result is consistent with the calculated residence time (τ). We have evaluated residence times for Li⁺ with respect to DMF and FSI⁻ (**Supplementary Fig. 15**). We observe a notable reduction of 60% in the residence time for both Li⁺/DMF and Li⁺/FSI⁻ pairs within LPE@Ni-DMF compared to pristine LPE. Furthermore, we can use this quantity to convert from residence time to the diffusion length of DMF co-migrating with Li⁺, calculated as $L = \sqrt{6D_{DMF}\tau}$. As shown in **Supplementary Fig. 17**, the diffusion coefficients for DMF (D_{DMF}) and FSI⁻ (D_{FSI^-}) remain relatively constant for both LPE and LPE@Ni-DMF. Therefore, both the residence time (τ) and diffusion length (L) of DMF moving together with Li⁺ are significantly reduced, which further elucidates the mitigating effect on DMF shuttling within LPE@Ni-DMF.

Moreover, the composition and stability of the solid-electrolyte interphase (SEI) formed on the lithium metal surface are also influenced by the presence of Ni-DMF in the LPE matrix, which is demonstrated both experimentally and computationally. Cryogenic transmission electron microscopy results show that the SEI formed in the presence of Ni-DMF is more stable, with more inorganic components and a higher fluorine-to-carbon (F/C) elemental ratio (**Fig. 4d-f**). The SEI characterization indicates that LPE@Ni-DMF is less prone to the adverse effects of DMF decomposition, as evidenced by the reduced weight loss after cycling compared to LPE. The COMSOL simulations present the kinetic process of DMF decomposition and SEI evolution during cycling process, which demonstrates the inhibition effect of DMF decomposition in LPE@Ni-DMF (**Fig. 4g and 4h**).

Accordingly, we added the statement “The diffusion length of DMF co-migrating with Li⁺ can be calculated by the equation of $L = \sqrt{6D_{DMF}\tau}$. As shown in Supplementary Fig. 17, the diffusion coefficients for DMF and FSI⁻ remain relatively constant in both LPE and LPE@Ni-DMF, while the diffusion coefficient for Li⁺ is higher in the latter. This significantly reduces the diffusion length of DMF moving together with Li⁺, which mitigates DMF shuttling within LPE@Ni-DMF.” on **Page 12**, and “In contrast, the dominant peak of the Li-O(DMF) pair in LPE@Ni-DMF suggests that

more DMF molecules participate in Li^+ solvation, resulting in a larger CN of 3.8 and a larger Li-O(DMF) distance of 2.11 Å in LPE@Ni-DMF compared to that in LPE (2.05 Å) (Fig. 2f and 2g). The DMF ligands of Ni-DMF alter the Li^+ coordination environment in LPE, leading to an increase in the number of solvent molecules surrounding Li^+ and consequently a longer Li-O(DMF) bond length. This reduces the interaction between Li^+ and DMF molecules, including those trapped in the PVDF-HFP matrix.” on Page 13 to the revised manuscript.

Fig. 2e and 2f | RDFs of Li^+ /DMF and Li^+ / FSI^- pairs in LPE (e) and LPE@Ni-DMF (f).

Supplementary Fig. 17 | The diffusion behavior of Li^+ , DMF, and FSI^- in LPE and LPE@Ni-DMF. (a) Mean square displacements (MSD), and (b) self-diffusion coefficients (D).

(a-2-2) Additional information that would be helpful to support the authors' claim

- If Ni-DMF can absorb extra DMF molecules in its crystalline structure (shown in Fig. 1a) or not
- o If yes, please state this in the manuscript with additional information about the crystalline structure of Ni-DMF with excess DMF molecules (somewhat similar to Supplementary Fig. 1a)
- Regardless of the answer to the above comment, the authors would need to improve the preparation method of LPE@Ni-DMF because it does not look like a straightforward method to evaluate the solvent-trapping ability of Ni-DMF when mixed with the DMF-containing LPE. The issue is that all ingredients were mixed in DMF, which might have allowed solvent DMF molecules to fill up all the adsorption sites in Ni-DMF. Therefore, the DMF content in the LPE matrix in LPE@Ni-DMF and that in the reference LPE (per unit volume of the LPE) would be the same as each other. To highlight the significance of the authors' Hofmann frameworks in mitigating DMF decomposition

on Li, the following experiments are recommended:

- o Mixing a melted LPE (<150 °C so as not to remove residual DMF molecules in the matrix) with Ni-DMF or Ni-activated and coating the mixture to obtain LPE@Ni-DMF or LPE@Ni-activated without any concerns about the presence of excess DMF molecules in the Hofmann framework
- o Inserting a Ni-DMF or Ni-activated thin layer between the LPE and Li and performing cell tests. The Ni-DMF or Ni-activated thin layer might be obtained by vacuum filtration of a Ni-DMF- or Ni-activated-containing solution or drop-casting of a Ni-DMF- or Ni-activated-containing solution on Li or the LPE followed by solvent evaporation.

Response:

Thanks for the reviewer's suggestions. With regards to the first question, we reanalyzed the fresh and cycled diffraction patterns of LPE@Ni-DMF extracted from 2D wide-angle X-ray scattering. It is clear that the diffraction peak assigned to the (200) crystalline phase of Ni-DMF exhibits a negligible shift compared to the pristine state of LPE@Ni-DMF, indicating Ni-DMF cannot absorb extra DMF molecules in its crystalline structure (**Fig. R4**).

Fig. R4 | Magnified diffraction patterns of LPE@Ni-DMF extracted from 2D wide-angle X-ray scattering before and after cycling.

To confirm the significant role of the Hofmann frameworks in mitigating DMF decomposition on Li, we conducted additional experiments as recommended by the reviewer. During the preparation of the composite electrolyte slurry, we replaced the DMF solvent with tetrahydrofuran (THF) while keeping all other conditions unchanged, due to its lower boiling point compared to DMF. The slurry containing Ni-DMF was then coated onto a Li plate, which was used as a composite anode (denoted as CA@Ni-DMF; **Fig. R5a**) after the THF solvent completely evaporated at room temperature. A reference sample was also prepared under the same conditions without Ni-DMF (denoted as CA; **Fig. R5b**). Full cells were assembled with SPAN as cathodes, LPE as solid electrolytes, and coated Li plates as composite anodes, respectively. The cycling performance of the cells were then tested at 0.2 C under room temperature. As shown in **Fig. R6a**, the discharge-charge curves of the SPAN/LPE/CA@Ni-DMF cell are substantially overlapped from the 1st to 20th cycles with negligible hysteresis aggravation, indicating excellent electrochemical reversibility. This result also demonstrates that the coating layer can induce a stable SEI to prevent DMF migration. In contrast, the SPAN/LPE/CA cell experiences significant micro-short circuiting during the charging process after the first cycle (**Fig. R6b**). This can be explained by the weaker ability of PVDF in Li salts dissociation compared to Ni-DMF, which induces an unstable SEI or even breakage of the coating

layer. As a result, DMF migration cannot be prevented, leading to direct decomposition on Li metal.

Fig. R5 | Digital photos of CA@Ni-DMF (a) and CA (b).

Fig. R6 | Discharge-charge curves of the SPAN/LPE/CA@Ni-DMF (a) and SPAN/LPE/CA (b) cells at 0.2 C.

(a-3) What are the electrochemical potential windows of Ni-H₂O, Ni-activated, Ni-DMF, LPE, and LPE@Ni-DMF?

The electrochemical potential window is one of the important pieces of information about solid-electrolyte membranes, which allows battery scientists to decide the working potential range of batteries, but no such information has been described in the manuscript. It seems the authors have tried to include this in the manuscript based on the description for an unused linear sweep voltammetry (Lines 597–599). It would be nice if the authors could include this information in the manuscript.

Response:

Many thanks for the reviewer's suggestion. We apologize for the missing information regarding the linear sweep voltammetry (LSV) data. The electrochemical potential window of electrolytes is an important criterion for evaluating the stability and practical application capability within the working potential range. In response to the suggestion, we conducted LSV tests on Ni-H₂O powders, Ni-activated powders, Ni-DMF powders, LPE membrane, and LPE@Ni-DMF membrane.

A tablet die was used to prepare a tablet for each powder sample using a press machine under 20 MPa (Fig. R7). Asymmetric SS/tablet/Li cells were assembled to conduct LSV in the potential range

of 0 to 5 V at a scanning rate of 1 mV s^{-1} . It is evident that no obvious redox potential can be detected for all tablet samples in the selected potential range, indicating the Hofmann framework materials are electrochemically stable.

For both membrane samples, we investigated their high voltage stability using asymmetric SS/membrane/Li cells (**Fig. 3g**). The LPE exhibits a sharp oxidizing peak beginning at approximately 4.3 V, while the LPE@Ni-DMF begins to oxidize at over 4.7 V, indicating a high antioxidative capability of LPE@Ni-DMF.

Accordingly, we added the statement “The electrochemical potential window of electrolytes is an important criterion for evaluating practicality. The high voltage stability of LPE and LPE@Ni-DMF was tested by linear sweep voltammetry (LSV). The LPE exhibits a sharp oxidizing peak beginning at approximately 4.3 V, while the LPE@Ni-DMF begins to oxidize at over 4.7 V, indicating a high antioxidative capability of LPE@Ni-DMF” to the revised manuscript. (**Page 16**)

Fig. R7 | Tablet die for Hofmann tablet samples preparation (a) and LSV data of Ni-H₂O tablet (b), Ni-activated tablet (c), and Ni-DMF tablet (d) with a scanning rate of 1 mV s^{-1} .

Fig. 3g | LSV curves of asymmetric SS//Li cells with a scanning rate of 1 mV s⁻¹.

(a-4) How do the Ni species in the authors' Hofmann frameworks affect the electrochemical reactions on the Li side (as well as the cathode side)?

It has been known that the transition-metal ions adversely affect the electrochemical reactions on the anode side [Adv. Energy Mater. 11 (2021) 2103005]. For example, transition metals of cathode active materials (e.g., LiNi_xCo_yMn_zO₂) would dissolve in an electrolyte solution (as cation forms), migrate to the anode, and participate in electrochemical reactions there. Reduced transition-metal ions would form metal nanoparticles on the anode, which acts as catalysts to promote lithium dendrite growth. With this in mind, it is of crucial importance to investigate whether Ni species in the authors' Hofmann frameworks adversely affect their cell performances or not. Since the Ni sources, i.e., NiCl₂·6H₂O and K₂[Ni(CN)₄], are originally small complexes that can dissolve in a solvent and the distance between the closest Ni-DMF in LPE@Ni-DMF and the Li-anode surface in the authors' experiments is not long, I believe there is a high likelihood in observing some sort of morphological, spectroscopical, and/or electrochemical footprints stemming from the Ni species in Ni-DMF.

Some recommended experiments are SEM, EDS, and XPS of the Li surface after symmetric or asymmetric cell cycling to see if Ni species can be detected. If not detected, such results should be described in the manuscript to relieve the above concern.

Response:

Many thanks for the reviewer's suggestion. We have carefully read the aforementioned reference [Adv. Energy Mater. 11 (2021) 2103005] and believe the issue of transition metal dissolution is more likely to occur in liquid-electrolyte based Li-ion batteries with layered LiNi_xCo_yMn_zO₂ as the cathode. During repeated discharging/charging processes at high voltage (approximately 4.2 or 4.3 V), the transition metal, especially Mn ion, undergoes a disproportionated reaction, leading to serious lattice distortion of the layered structure and Mn ion dissolution.

In our solid-state Li-sulfur batteries, the working potential ranges from 1 to 3 V, and the single Ni species are electrochemically stable at 3 V. Furthermore, Ni-DMF materials were synthesized through a ligand-exchange process using DMF as the ligand. This ensures that there is no possibility of Ni-DMF dissolving in the residual DMF solvent of LPE@Ni-DMF.

SEM was implemented to analyze the morphology of the Li metal surface after Li/LPE@Ni-DMF/Li symmetric cell testing at room temperature (20 hours of cycling at 0.1 mA cm⁻² with capacity of 0.1 mAh cm⁻² per half cycle). EDS and XPS were used to detect Ni species on the Li metal surface. As shown in **Supplementary Fig. 35**, the cycled Li metal exhibits smooth surface morphology without any Ni metal nanoparticles or other detectable sources, which is further confirmed by the EDS mapping data.

Supplementary Fig. 35 | (a) SEM image of the Li metal surface after Li/LPE@Ni-DMF/Li symmetric cell testing at room temperature (20 hours of cycling at 0.1 mA cm^{-2} with capacity of 0.1 mAh cm^{-2} per half cycle) and (b) corresponding EDS mappings.

XPS was used to detect the presence of Ni on the surface of the cycled Li metal. It should be noted that the cycled Li metal was washed three times with 1,2-Dimethoxyethane (purity: $> 99.9 \%$, DoDoChem) solvent and then dried under vacuum for 6 hours at room temperature before the XPS testing. The Ni2p XPS spectrum (**Supplementary Fig. 38**) show that no Ni signal can be detected, suggesting that the Ni-DMF framework structure is relatively stable during cycling, which is consistent with the EDS mapping results.

Supplementary Fig. 38 | Ni2p XPS spectrum of the Li metal surface after Li/LPE@Ni-DMF/Li symmetric cell testing at room temperature (20 hours of cycling at 0.1 mA cm^{-2} with capacity of 0.1 mAh cm^{-2} per half cycle).

Similarly, we conducted SEM and EDS to determine if the Ni species in the Hofmann frameworks affect the SPAN cathode side. No Ni trace was found in the cycled SPAN electrode based on SEM and corresponding EDS mapping data (**Fig. R8**). In summary, we conclude that the Ni species in our Hofmann frameworks do not affect the electrochemical reactions on either the Li or the cathode sides.

Fig. R8 | (a) SEM image of the surface of SPAN cathode after SPAN/LPE@Ni-DMF/Li cell testing at room temperature (10 cycles at 0.2 C) and (b) corresponding EDS mappings.

Accordingly, we added the statement “It should be noted that the SEM and corresponding EDS mappings show that the Li metal after Li/LPE@Ni-DMF/Li cell cycling has a smooth surface morphology without any detectable Ni species, indicating the good electrochemical stability of Ni-DMF on Li metal (Supplementary Fig. 35)” on Page 20, and “The Ni_{2p} XPS spectrum (Supplementary Fig. 38) indicates that no Ni signal is present in the SEI composition, which is consistent with the EDS mapping results” on Page 21 to the revised manuscript.

(a-5) How does the incorporation of Ni-DMF into the LPE alter the ion-conduction behavior of the interface between Ni-DMF and the LPE?

In the manuscript, it seems not enough information has been stated for this point (except for Supplementary Fig. 26). However, many reports state the reduction of the crystallinity of a polymer attached to the surface of a material doped in its matrix, thereby improving the ionic conductivity of the composite material [e.g., Ref. 13, Energy Environ. Sci. 13 (2020) 2386–2403]. Importantly, some dielectric ceramic materials are known to induce the dehydrofluorination of PVDF as evidenced by the formation of C=C double bonds (e.g., Ref. 13), which alters the ion-conduction behavior at the interface between the doped material and PVDF.

Therefore, the authors are requested to provide more information about the ion-conduction behavior at the interface between Ni-DMF and the LPE (e.g., differential scanning calorimetry to see the shift of glass transition temperature of PVDF-HFP and/or FTIR to see the formation of C=C bonds in PVDF after Ni-DMF addition) and need to specify the multiple conduction pathways in LPE@Ni-DMF (i.e., through the LPE matrix, through Ni-DMF, and through the LPE|Ni-DMF interface). In this context, it would be valuable if the authors could measure the ionic conductivity of a Ni-DMF thin layer and/or Ni-DMF with LiFSI as this can provide useful information about which ion-conduction pathway is the most conductive and if there is a synergistic effect of mixing Ni-DMF and the LPE on conductivity enhancement.

Response:

Many thanks for the reviewer's suggestion. We apologize for the inadequate information and explanation regarding the improved ion-conduction behavior at the interface between Ni-DMF and the LPE. Obtaining accurate data on the glass transition temperature for LPE or LPE@Ni-DMF through differential scanning calorimetry testing is challenging due to the evaporation of the solvent in DMF-containing solid electrolytes during the temperature-rise period of the testing. This affects the synchronous collection of heat flow and, therefore, the accuracy of the data. However, we agree

with the reviewer that the formation of C=C double bonds, after the dehydrofluorination of PVDF, can alter the ion-conduction behavior at the interface between the Ni-DMF and the polymer matrix. The ATR-FTIR tests of LPE and LPE@Ni-DMF electrolyte membranes were conducted to evaluate the trace of C=C double bonds (**Supplementary Fig. 24**). It is evident that the spectrum of LPE@Ni-DMF exhibits a more pronounced C=C characteristic peak at 1703 cm^{-1} than LPE, demonstrating that the Ni-DMF can induce improved dehydrofluorination of PVDF-HFP, thereby improving the ion-conduction at the interface of the composite material.

Supplementary Fig. 24 | ATR-FTIR spectra of LPE@Ni-DMF (a) and LPE (b).

To further evaluate the significance of Ni-DMF in the composite material interface, we conducted Raman tests for both LPE and LPE@Ni-DMF. The S-N-S band of FSI^- in the Raman spectra consists of four modes: free FSI^- , contact ion pairs (CIP), and aggregate clusters (AGGs). From **Fig. 3e and 3f**, the content of free FSI^- (723 cm^{-1}), CIP (732 cm^{-1}), and AGGs of LPE are 6.4%, 24.7%, and 68.9%, respectively. In the case of the LPE@Ni-DMF, the free FSI^- content increases to 31.7%, while the proportions of CIP and AGGs decrease significantly, suggesting that solvation structures can be changed by the addition of Ni-DMF and hence the formation of more mobile Li^+ at the interfaces, which is consistent with the FTIR spectra results.

Fig. 3e and 3f | Raman spectra of LPE (e) and LPE@Ni-DMF (f).

Accordingly, we added the statement “ATR-FTIR tests of LPE and LPE@Ni-DMF membranes were performed to evaluate the trace of C=C double bonds⁹. It is evident that the spectrum of LPE@Ni-DMF shows a more prominent C=C characteristic peak at 1703 cm^{-1} than that of LPE, demonstrating that the Ni-DMF can induce enhanced dehydrofluorination of PVDF-HFP, thereby

improving the ionic conduction at the interface of the composite material (Supplementary Fig. 24). Raman analysis was performed to detect the FSI⁻ anion states¹⁵. From Fig. 3 e and f, the content of free FSI⁻, contact ion pairs (CIP), and aggregate clusters (AGGs) of LPE are 6.4%, 24.7%, and 68.9%, respectively, while the corresponding values for LPE@Ni-DMF are 31.7%, 21.1%, and 47.2%, respectively, suggesting that the solvation structures can be altered by the addition of Ni-DMF and hence the formation of more mobile Li⁺.” to the revised manuscript. (Page 16)

To confirm the positive effect of Ni-DMF in ion-conduction, we conducted additional experiments as recommended by the reviewer. During the preparation of the electrolyte membranes, we replaced the DMF solvent with THF while keeping all other conditions unchanged (Fig. R9). EIS test was performed to obtain the ionic conductivities of solvent-free composite electrolytes (Fig. R10). The LPE/THF exhibits a lower ionic conductivity of $8.8 \times 10^{-6} \text{ S cm}^{-1}$ than that of LPE@Ni-DMF/THF ($1.5 \times 10^{-5} \text{ S cm}^{-1}$), suggesting the formation of more movable Li⁺ owing to the notable dissociation of LiFSI activated by Ni-DMF. Therefore, the use of DMF as a plasticizer is effective in improving the ion-conduction of the SPE, as stated in our manuscript. However, the combination of Ni-DMF and LPE has a synergistic effect that enhances conductivity and reduces the adverse impact of the solvent in SPE. We hope that our explanations have addressed your concerns adequately.

Fig. R9 | Digital photos of LPE/THF (a) and LPE@Ni-DMF/THF (b) membranes.

Fig. R10 | Nyquist plots of LPE/THF (a) and LPE@Ni-DMF/THF (b) at room temperature.

The authors are required to address the aforementioned concerns and revise the manuscript to tell a clear story about the significance of their results in using Hofmann frameworks for solid-state batteries.

Response:

We have carefully replied the reviewer's questions and revised our manuscript to emphasize the novelty and significance of our results in using Hofmann frameworks for solid-state batteries. We hope that these responses address your concerns satisfactorily.

Although revising the story is the top priority, the following point-by-point suggestions would help improve the manuscript as well:

[b. Major comments]

(b-1) Lines 38-39 and 530-531: Please add the time for half cycles (either "1 h" or "0.1 mAh cm⁻²"). This is because the cycle performance depends on how much Li is deposited/stripped per half cycle, which is very important information.

Response:

Many thanks for the reviewer's suggestion. We apologize for the lack of information regarding the Li//Li symmetric cell test. Accordingly, the relevant data have been added to the revised manuscript as described below:

"The locally solvent-tethered electrolyte stably cycled for over 6,000 hours at 0.1 mA cm⁻² with a repeated plating/stripping capacity of 0.1 mAh cm⁻² in Li//Li symmetric cells." (Page 2) and "Our results demonstrate that the confinement effect of Ni-DMF prevents DMF consumption, resulting in enhanced durability for both the CPE and Li metal anode (over 6,000 hours of cycling at 0.1 mA cm⁻² with capacity of 0.1 mAh cm⁻² per half cycle)." (Page 30)

(b-2) Lines 79-81: Please revise this sentence because, as I stated in (a-1-2), this appears not to be true.

Response:

We acknowledge the reviewer's careful review. We have rewritten the sentence to point out the dilemma more clearly in polymer electrolyte design as described below:

"However, previous approaches to additive engineering have mainly focused on addressing either ionic conductivity or electrochemical stability. The synergetic effect of DMF modulation and Li⁺ transport mechanism among the complex components remains unclear, which substantially impedes the design of high-performance and durable polymer electrolytes. In brief, an effective strategy to simultaneously improve both aspects needs to be developed for the ultimate application of SPEs." (Page 4)

(b-3) Lines 222-224: Although this sentence would be true, when we see the trajectory of the focused Li⁺ ion in Fig. 2a, it just vertically moved close to the Ni-DMF surface, which means this Li⁺ ion did not contribute to the Li⁺-ion-conduction through LPE@Ni-DMF (from left to right or vice versa in the figure). It would be nice if the authors could deliver the same conclusion using a different Li⁺-DMF pair.

Response:

We thank the reviewer for the suggestion. Indeed, ionic conduction behaviors of Li⁺ in vertical and

parallel orientations to the interface are both important. Therefore, we have extracted snapshots from the MD trajectories for a different Li⁺-DMF pair (Li atom ID: 8119) (**Supplementary Fig. 13**). These additional snapshots elucidate not only the horizontal translational motion of Li⁺ but also the rotational dynamics of the DMF ligands.

Supplementary Fig. 13 | Snapshots of MD trajectories showing local motion of the DMF ligand during Li⁺ transfer.

(b–4) Lines 232–233 and 749–751: Did the molecular dynamics simulation contain PVDF-HFP? Figs. 2a–2c and Supplementary Figs. 12 and 13 do not clearly show the contents related to PVDF-HFP. If it has been removed from the simulation, it would mean the authors' computational result might not provide a reasonable deduction of the ion-conduction mechanisms in the real LPE and LPE@Ni-DMF.

Response:

We thank the reviewer for this thoughtful question. In this work, we primarily focus on the role of the DMF ligand-exchanged Hofmann frameworks, which greatly enhances both the ionic conductivity and stability of the SEI. Therefore, the PVDF-HFP phase was not incorporated in our theoretical framework. There are two reasons for adopting a simplified model for the composite electrolyte. First, the influence of PVDF-HFP on Li⁺ transport is comparatively marginal in comparison to the Ni-DMF additive. As evidenced by an earlier study conducted by Chunsheng Wang and co-workers, PVDF-HFP primarily serves as an inert support matrix with polymer-based electrolytes, playing a limited role in the ionic conduction of Li⁺ (*Nat. Energy* (2024) doi: 10.1038/s41560-023-01443-0). Second, there are numerous challenges associated with the accurate description of molecular interactions involving polymers within a highly ionic environment. Factors such as the degree of polymerization and the molecular weight distribution contribute significantly to the complexity of polymer behavior (*Phys. Rev. Lett.* **98**, 227802 (2007); *Macromolecules* **48**, 7882–7888 (2015); *Phys. Rev. Lett.* **110**, 018301 (2013); *Chem. Sci.* **10**, 8724–8734 (2019)). Additionally, PVDF is a semicrystalline polymer that can crystallize in five different phases that are correlated to distinct chain conformations and motion modes (*Nat. Protoc.* **13**, 681–704 (2018)). Therefore, the accurate simulation of polymers requires a substantially larger spatial scale of the theoretical model, which exceeds the scope of our effort aimed at obtaining mechanistic insights into the interactions between local DMF-ligand and Li⁺. Given the relatively minimal impact of polymers on Li⁺ transport and the inherent modeling challenges, we opted against incorporating a full polymer phase in our investigation. We hope that our explanations have addressed your concerns adequately.

(b-5) Fig. 2h (Lines 267–268): I am not sure if this computational result is truly informative because, in the real Ni-DMF, the lattice spacing was identified to be 7.63 Å (Fig. 1a), whereas this simulation employed nearly 8 times higher lattice spacing (i.e., 60 Å). Therefore, it is unknown whether the same phenomenon happened in the real Ni-DMF or not.

Response:

Thanks for the reviewer's comment. We apologize for any confusion our description may have caused. There appears to be a misunderstanding here, and the simulation in **Fig. 2h** shows the probability distribution of DMF and FSI⁻ between two separate Ni-DMF nanosheets, not within the interlayer of a single nanosheet. This result reflects the varying species distribution across the bulk LPE phase and the interface of Ni-DMF. The sentence has been corrected in the revised manuscript as described below:

“Probability distribution of DMF and FSI⁻ between two separate Ni-DMF nanosheets in LPE@Ni-DMF.” (**Page 14**)

(b-6) Lines 293–294: The current rate [mA/cm²], plating/stripping amount [mAh/cm²], and temperature need to be provided.

Response:

Thanks for the reviewer's suggestion. The detailed test information has been added to the revised manuscript (**Page 16**) as described below:

“Furthermore, a significant difference is observed between cycled samples (galvanostatic plating/stripping tests at 0.1 mA cm⁻² and 0.1 mAh cm⁻² per half cycle for 100 hours in Li//Li symmetric cells at room temperature), where...”.

(b-7) Supplementary Figs. 21 and 22 (Lines 300–301): How were these experiments performed? It seems no information can be found in the manuscript.

Response:

Thank you for pointing this out, and we apologize for the confusion. We have added the experimental details regarding in-situ optical microscope test in the revised manuscript (“Materials characterization” section, **Page 34**) as described below:

“The in-situ reactivity between DMF and Li metal was investigated using an optical microscope (DVM6). The lithium foil was cut to a size of 0.5 × 1 cm² and then placed in the porthole of a Teflon mold (**Fig. R11**). The porthole was then filled with DMF solvent and sealed with a glass slide. All operations were performed in an Ar-filled glovebox.”

Fig. R11 | Teflon mold for light microscope investigation.

(b–8) Supplementary Fig. 26 (Lines 323–326): How can the authors ensure this sentence, although the diffraction peaks of PVDF-HFP appear at around the same 2θ values as Ni-DMF? Is there no possibility of overlapping the diffraction peaks of both PVDF-HFP and Ni-DMF?

Response:

We appreciate the reviewer's thoughtful comment and apologize for any confusion. It is true that the diffraction peaks of PVDF-HFP and Ni-DMF can overlap at the same 2θ values. Therefore, we conducted additional XRD tests on LPE@Ni-DMF before and after cycling to detect the full width at half-maximum (FWHM) of the amorphous area (**Supplementary Fig. 31**). It is clear that both samples exhibit similar FWHM, indicating the predominant amorphous structure in LPE@Ni-DMF can be well-preserved after cycling.

Supplementary Fig. 31 | XRD patterns of LPE@Ni-DMF before and after cycling.

Accordingly, we reorganized the sentence and added the statement “Additionally, the XRD pattern of cycled LPE@Ni-DMF shows a similar full width at half-maximum of the amorphous area as that of fresh LPE@Ni-DMF, indicating the predominant amorphous structure in LPE@Ni-DMF is well maintained” to the revised manuscript. (Page 18)

(b–9) Supplementary Fig. 29: Why the high-frequency intercepts of both Nyquist plots are the same as each other? If LPE@Ni-DMF is more ion-conductive than the LPE, its Nyquist plots should be

shifted to the left. Is there a difference in thickness between LPE@Ni-DMF and the LPE?

Response:

Many thanks for the reviewer's comment. The high-frequency intercept indicates the ohmic resistance, which is mainly attributed to the electrolyte and electrical connection resistances, while the semicircle indicates the interphase resistance. In the case of LPE, the uncontrolled decomposition and continuous depletion of DMF results in a thick and unstable SEI between the electrolyte and the electrode, which features an enlarged semicircle compared to that of LPE@Ni-DMF. The similar high-frequency intercepts may be attributed to the decomposition of DMF, which causes a change in the thickness of the solid electrolyte. Although the conductivity decreases, the thickness becomes thinner, resulting in a similar ohmic resistance.

(b-10) Lines 378–379: The phrase “with a concentrated distribution of C, N, O, F, and S elements” needs to be revised because LPE@Ni-DMF also shows the concentrated distribution of these elements in particular regions in Supplementary Fig. 33.

Response:

Many thanks for the reviewer's suggestion. The corresponding sentences have been rewritten in the revised manuscript (**Page 21**) as described below:

“Although the distribution of C, N, O, F, and S elements is concentrated in both LPE-SEI and LPE@Ni-DMF-SEI, there are notable distinctions in the induced depositional morphology. LPE@Ni-DMF-SEI has a compact and thinner morphology compared to LPE-SEI, which exhibits an uneven deposition.”

(b-11) Supplementary Fig. 33: It seems the area observed for the TEM image in (a) is different from that used for element mappings in (b). Please check it and update the figure.

Response:

We express our gratitude to the reviewer's careful examination of our work, and we apologize for the oversight. The TEM image and corresponding element mappings have been updated and listed in the revised Supplementary information as **Supplementary Fig. 40**.

Supplementary Fig. 40 | Cryogenic transmission electron microscopy image (a) and corresponding element mappings (b) of LPE@Ni-DMF-SEI.

(b-12) Line 421: Was Fig. 4c measured by CV? If so, please specify the experimental condition carefully (e.g., voltage range, starting voltage, rest time, and if it is the two-electrode measurement,

how did the authors define 0 V vs. Li/Li+).

Response:

Many thanks for the reviewer's comments and we apologize for the incomplete experimental condition. Tafel curves were extracted from CV test and the experimental details have been added to the revised manuscript ("Electrochemical measurements" section, **Page 33**) as described below: "Cyclic voltammetry (CV) measurements of the Li/SSE/Li cells were performed on an electrochemical workstation at initial open-circuit voltage over the voltage range of $-0.2 \sim 0.2$ V with a scanning rate of 0.1 mV s^{-1} . Tafel curves were obtained by fitting the second cycle (after SEI formation) of CV data including anodic scan and cathodic scan. The voltage at the intersection of two curves was defined as 0 V (around the Li/Li⁺ equilibrium potential)."

Additionally, the caption of **Fig. 4c** was revised to "Tafel curves with calculated ECD of the Li//Li symmetric cells." (**Page 24**)

(b-13) Lines 440-442: The theoretical capacity of SPAN needs to be specified somewhere around here. In addition, please make sure to explain the definition of 1C.

Response:

Many thanks for the reviewer's suggestion and we apologize for any confusion. Sulfurized polyacrylonitrile (SPAN) is a product of the polymerization of acrylonitrile monomer with sulfur, and usually, we take the theoretical capacity of the active component sulfur (1675 mAh g^{-1}) as the specific capacity of the SPAN system. 1 C indicates that the cells are rated at the one-hour rate. For example, if a SPAN cathode with 0.5 mg sulfur is cycled at 1 C, the current is calculated to be 0.8375 mA. We have added the theoretical capacity of SPAN to the revised manuscript (**Page 25**) to make a clearer illustration as described below:

"SPAN cathodes (high theoretical capacity of 1675 mAh g^{-1}) and Li metal anodes were paired for both LPE and LPE@Ni-DMF to assemble SPAN/LPE/Li and SPAN/LPE@Ni-DMF/Li coin cells and were tested at multiple temperatures."

(b-14) Lines 447-451: The authors need to provide the charge-discharge curves at 150th-200th cycles (e.g., every 10 cycles) in the Supporting Information and to explain why they thought the severe capacity drop was caused by micro-short circuiting by pointing out a characteristic voltage-current response stemming from it. Additionally, dQ/dV analysis would help identify some differences between the voltage-current responses easily.

Response:

Many thanks for the reviewer's suggestions. The charge-discharge curves at 0.2 C and corresponding dQ/dV analysis of both SPAN/LPE/Li and SPAN/LPE@Ni-DMF/Li cells have been added to the revised Supplementary information as **Supplementary Fig. 47**. Meanwhile, we have reasonably explained the cause of the severe capacity drop as micro-short circuiting as described below:

"As one of the most common failures of solid-state batteries, "short circuit" can be categorized as "hard short circuit" or "soft short circuit" (also known as micro-short circuit). In the case of a "hard short", the voltage drops dramatically during the charging process and the battery is unable to recover, which is the most common short circuit phenomenon. In contrast, the "soft short" phenomenon is often observed in solid-state batteries, where the battery voltage is dynamically fluctuated but does not rise during charging, and the battery is able to recover from the short circuit.

The voltage fluctuation is mainly caused by an uneven local electric field distribution due to lithium dendrite growth. This results in a decrease in Coulombic efficiency and a subsequent severe drop in capacity. As demonstrated in **Supplementary Fig. 47a**, the phenomenon of voltage fluctuation is evident during the charging process, particularly for the 150th, 190th, and 200th cycles. This is further supported by the dQ/dV curves, which clearly show a significant raise or drop in potential. In contrast, the phenomenon of voltage fluctuation is not apparent in the SPAN/LPE@Ni-DMF/Li cell (**Supplementary Fig. 47b**), which demonstrates that our strategy can create a stable interface between LPE@Ni-DMF and Li metal, preventing micro-short circuits.”

Supplementary Fig. 47 | Charge–discharge curves of SPAN/LPE/Li (a) and SPAN/LPE@Ni-DMF/Li (b) cells at the 150th, 160th, 170th, 180th, 190th, and 200th cycles at 0.2 C. The figures inserted depict the corresponding dQ/dV results.

(b–15) Lines 450–451: This sentence needs a revision because the capacity fading behaviors in Fig. 5b are not the same as those in Fig. 5a. The obvious fact is that LPE@Ni-DMF showed a better capacity retention than the LPE at both 0.2C and 1C. It would be nice if the authors could revise the sentence to improve readability.

Response:

Many thanks for the reviewer's suggestions and we have reorganized the expression in the revised manuscript (**Page 26**) to improve readability as described below:

“The improved electrochemical performance of the SPAN/LPE@Ni-DMF/Li cell is also validated by cycling at 1 C (Fig. 5b). The SPAN/LPE@Ni-DMF/Li cell displays remarkable cyclability (1,000 cycles with a capacity retention of 60%) and records an average CE of 99.8% throughout its cycle life. In contrast, the SPAN/LPE/Li cell fails after only 100 cycles with a low average CE of 97.1%.”

(b–16) Fig. 5d (Lines 481–483): For a fair comparison, the top half of the y-axis in Fig. 5d needs to be (retained) areal capacity [mAh/cm²], which can include the difference in the sulfur loading between the studies. If the authors can, making 3D plots of this dataset using the x-axis of (retained) energy density [Wh/kg-cathode active material], the y-axis of (retained) power density [W/kg-cathode active material], and the z-axis of the maximum cycle number would identify the best performance more clearly.

Response:

Many thanks for the reviewer's suggestions and we do agree with the reviewer that the areal capacity is a key parameter to evaluate Li-S cells performance. The comparison of areal capacity between

references and our work has been updated in the revised manuscript (**Fig. 5d**). The areal capacities of LPE@Ni-DMF based cells are comparable to that of other studies. Additionally, we also made a 3D plot to exhibit performance comparison in terms of energy density, power density, and maximum cycle number. It should be noted here the calculated results are based on the mass of cathode active material (sulfur). The average cell voltage is 1.9 V for SPAN cathode and 2.1 V for sulfur cathode, respectively. As shown in **Fig. R12**, our cells deliver the best energy density of 1826 Wh kg⁻¹ after 300 cycles at 0.2 C and the best power density of 950 W kg⁻¹ after 1000 cycles at 1 C.

Fig. 5d | Comprehensive comparison of polymer-based ssLSBs performance (including specific capacity, areal capacity, and cyclability) between this work and those in previous reports.

Fig. R12 | Comprehensive comparison of polymer-based ssLSBs performance (including energy density, power density, and maximum cycle number) between this work and those in previous reports.

(b-17) Lines 558–559: This procedure needs to be explained more clearly. For instance, what were the weights of a DMF solvent and Ni-activated?

Response:

Many thanks for the reviewer's comment. We have supplemented the detailed information in the revised manuscript (“Materials preparation” section, **Page 31**) as described below:

“100 mg of Ni-activated powder was immersed in 2 mL of DMF (purity: > 99.8%, Aldrich) solution (~1.9 g), followed by magnetically stirring for 2 hours to fulfill thorough solution exchange process. After the color of Ni-activated thoroughly changed to light green, ...”

(b-18) Materials preparation: The supplier and grade information of each material should be provided. These materials are:

- NiCl₂·6H₂O
- K₂[Ni(CN)₄]
- DMF
- PVDF-HFP
- LiFSI
- SPAN
- Multi-walled carbon nanotubes
- Al foil (please include the information about its thickness)
- Li metal
- Al tab
- Ni tab
- Al compound packing film

Response:

Many thanks for the reviewer's suggestions. The supplier and grade information of materials have been added to the revised manuscript (“Materials preparation” and “Electrochemical measurements” sections) as highlighted below:

“Ni(H₂O)₂Ni[CN]₄·xH₂O was synthesized via a coprecipitation method. In a typical procedure, 0.19 g of NiCl₂·6H₂O (purity: > 99%, Macklin) and 0.176 g of dihydrate tri-sodium citrate (purity: > 99%, Aldrich) were dissolved in 40 mL of deionized water under stirring for 1 hour to form a clear solution. 0.193 g of K₂[Ni(CN)₄] (AR, Aldrich) was added into 40 mL deionized water to form another clear solution. After that, the two solutions were mixed under agitated stirring for 4 hours and then aged at room temperature for 24 hours. The light-blue precipitate was collected out and washed with deionized water three times denominated as Ni-H₂O. Followed by thermal dehydration treatment of Ni-H₂O under a vacuum condition at 80 °C for 6 hours, the light-yellow sample (Ni(H₂O)₂Ni[CN]₄) was achieved denominated as Ni-activated. 100 mg of Ni-activated powder was immersed in 2 mL DMF (purity: > 99.8%, Aldrich) solution (~1.9 g), followed by magnetically stirring for 2 hours to fulfill thorough solution exchange process. After the color of Ni-activated thoroughly changed to light-green, the residual DMF and H₂O molecules were eliminated by centrifugation and the light-green precipitate (Ni(DMF)₂Ni[CN]₄) underwent vacuum drying at 60 °C for 2 hours to yield the final product Ni-DMF.

The solid-state electrolytes (SSEs) membranes were prepared through a solution-casting method. 0.3 g of PVDF-HFP (M.w. ~400,000, Maclin) was dispersed in 1.5 mL of DMF solution (~1.42 g) at 80 °C for 2 hours. 0.3 g of LiFSI (purity: > 99.9%, DoDoChem) was dissolved in 1.5 mL of DMF

solution (~1.42 g) at room temperature for 1 hour. PVDF-HFP and LiFSI solutions were then mixed, followed by stirring at room temperature for 4 hours. Then the mixture was cast onto a glass substrate using a doctor blade with 750 μm height, and the DMF solution was evaporated at 60 $^{\circ}\text{C}$ for 4 hours to obtain the LPE membrane. With other conditions unchanged, 60 mg of Ni-DMF powder was added into the mixture of PVDF-HFP and LiFSI solution with another 2 hours mixing performed before the casting process (the weight ratios between Ni-DMF, PVDF-HFP, LiFSI, and DMF is measured to be 1:5:5:47), and the LPE@Ni-DMF membrane can be obtained.

The SPAN composite was prepared by mixing sulfur (Alfa Aesar) and polyacrylonitrile (Mw = 150,000, Aldrich) in a weight ratio of 4:1, followed by heating treatment in an Ar-filled furnace at 450 $^{\circ}\text{C}$ for 6 h. The cathode slurry was firstly prepared by heat-dissolving 0.2 g of PVDF-HFP and 0.2 g of LiFSI in 5 mL of DMF solution at 80 $^{\circ}\text{C}$ for 4 hours. Then, 0.5 g of SPAN and 0.1 g of multi-walled carbon nanotubes (>95%, XFNANO) were mixed into the slurry. The electrodes were prepared by coating the slurry onto an Al foil (15 μm , Canrd) and dried at 60 $^{\circ}\text{C}$ for 12 hours. The sulfur loading of SPAN cathodes was calculated to be 1~2 mg cm^{-2} . All the preparation processes were carried out in an Ar-filled glovebox.

For the pouch cells assembly, SPAN coated on the Al current collector was used as the cathode (areal loading: 2 mg cm^{-2}), and lithium foil (100 μm) on the Cu current collector (9 μm , Canrd) was used as the anode. Both the anode and cathode were 5 \times 5 cm^2 while the solid-state electrolyte was 6 \times 6 cm^2 . It is noted that the Al current collector was welded to an Al tab (Canrd) and the Cu current collector was welded to a Ni tab (Canrd) with tab sizes of 0.1 \times 5 cm^2 . The whole electric core was finally packed with Al-plastic film (113 μm , Canrd) in a drying room with a humidity of less than 10% at room temperature.”

(b–19) Lines 562–568: This procedure needs to be explained more clearly. Please provide the following information:

- The gap between the doctor blade and the glass substrate
- For LPE@Ni-DMF:
 - o The weight ratios between Ni-DMF, PVDF-HFP, LiFSI, and DMF before casting
 - o The order of addition (e.g., weighed PVDF-HFP first, then LiFSI was added onto it followed by the addition of DMF. After stirring the mixture at 80 $^{\circ}\text{C}$ for 4 h, Ni-DMF was added and another 1 h mixing was performed)
 - o Mixing time after Ni-DMF addition

Response:

Many thanks for the reviewer's suggestions. We have provided more experimental details to describe the preparation process in the revised manuscript (“Materials preparation” section, **Page 31**) as highlighted below:

“The solid-state electrolytes (SSEs) membranes were prepared through a solution-casting method. 0.3 g of PVDF-HFP (M.w. ~400,000, Maclin) was dispersed in 1.5 mL of DMF solution (~1.42 g) at 80 $^{\circ}\text{C}$ for 2 hours. 0.3 g of LiFSI (purity: > 99.9 %, DoDoChem) was dissolved in 1.5 mL of DMF solution (~1.42 g) at room temperature for 1 hour. PVDF-HFP and LiFSI solutions were then mixed, followed by stirring at room temperature for 4 hours. Then the mixture was cast onto a glass substrate using a doctor blade with 750 μm height, and the DMF solution was evaporated at 60 $^{\circ}\text{C}$ for 4 hours to obtain the LPE membrane. With other conditions unchanged, 60 mg of Ni-DMF powder was added to the mixture of PVDF-HFP and LiFSI solution with another 2 hours mixing

performed before the casting process (the weight ratios between Ni-DMF, PVDF-HFP, LiFSI, and DMF were measured to be 1:5:5:47), and the LPE@Ni-DMF membrane can be obtained.”

(b–20) Lines 597–599: As I stated in (a–3), please include the LSV data of each material in the manuscript.

Response:

Many thanks for the reviewer's comment. We have supplemented the LSV data of LPE and LPE@Ni-DMF in **Fig. 3g** with corresponding descriptions in the revised manuscript (**Page 16**).

(b–21) Lines 614–617: The authors need to provide more information about the pouch-cell assembly. For instance:

- The size of each electrode
- The size of an LPE@Ni-DMF sheet
- The size of each tab
- The type of the tab used for each electrode (I presume an Al tab for SPAN/Al and a Ni tab for Li metal)
- How did the author attach a tab onto each electrode (used a tab welding machine?), especially for the Li metal anode?

Response:

We appreciate the reviewer's thorough feedback and apologize for any confusion caused by our unclear expression. We have included additional information and rewritten the procedure regarding the pouch-cell assembly in the revised manuscript (“Electrochemical measurements” section, **Page 33**) as described below:

“For the pouch cells assembly, SPAN coated on the Al current collector was used as the cathode (areal loading: 2 mg, cm⁻²), and lithium foil (100 μm) on the Cu current collector (9 μm, Canrd) was used as the anode. Both the anode and cathode were 5 × 5 cm² while the solid-state electrolyte was 6 × 6 cm². It is noted that the Al current collector was welded to an Al tab (Canrd) and the Cu current collector was welded to a Ni tab (Canrd) with tab sizes of 0.1 × 5 cm². The whole electric core was finally packed with Al-plastic film (113 μm, Canrd) in a drying room with a humidity of less than 10% at room temperature.”

(b–22) Line 637: The authors need to explain how the Li metal tablet was washed in detail.

Response:

Many thanks for the reviewer's comment. The detailed procedures for Li metal tablet cleaning have been added to the revised manuscript (**Page 34**) as described below:

“For XPS measurements, each Li tablet (after Li//Li cell cycling) was washed with 1,2-Dimethoxyethane (purity: > 99.9 %, DoDoChem) solvent for three times to remove the residual DMF on Li tablet. Then the washed Li tablet was dried under vacuum for 6 hours at room temperature before transferred and sealed into the XPS holder in the Ar-filled glovebox.”

(b–23) Line 727: This current density is exceptionally high compared to these applied in the actual experiments (100 vs. 0.1–1 mA cm⁻²). Is there any justification?

Response:

Many thanks for the reviewer's comment. We apologize for any confusion our simulation setup may

have caused. There seems to be a misunderstanding here, and indeed the current density referred to in this context is not the applied current density, nor is it the current density measured during routine testing. Rather, it is the exchange current density, which is specifically used to describe the ability to transfer charge across the electrode-electrolyte interface. Higher exchange current densities indicate faster charge transfer rates. The value for the exchange current density is a default reference value set in the COMSOL software, which we have not modified. Both the experimental and control samples were configured with the identical value. The same parameter can also be found in other literature (*PNAS* **120**, e2307847120 (2023)). In addition, there was an inadvertent error in the unit for this parameter; it should actually be “ A m^{-2} ” and not “ mA cm^{-2} ”, and we have already corrected this in the revised manuscript (**Page 38**).

(b–24) Supplementary Table 1: The reference for the first row might be 13 and that for PVBL might be 15. In the caption, the name of the table needs to be “Supplementary Table 1” (please remove “S” before “1”). In the footnotes, please explain all abbreviations (e.g., PMLSE, HSE-etched bmLLZO30, PHLC (20% CAP), etc.).

Response:

We express our gratitude to the reviewer’s careful examination of our work, and we apologize for the oversight. **Supplementary Table 1** has been updated with all abbreviations explained in the revised Supplementary information as outlined below:

Supplementary Table 1 | Comparison of ionic conductivity (σ) and Li^+ transference number (t_{Li^+}) between this work and those in previous reports.

Sample name	σ (S cm^{-1})	t_{Li^+}	ref.
PVDF-LiFSI ^a	1.18×10^{-4}	/	S4
PMLSE ^b	2.0×10^{-4}	0.62	S5
HSE-etched bmLLZO30 ^c	4.5×10^{-4}	/	S6
LATP-PVDF-Li ^d	2.44×10^{-4}	0.52	S7
PHLC (20% CAP) ^e	1.25×10^{-4}	0.49	S8
S-LHCE ^f	2.7×10^{-4}	0.72	S9
PVDF-DMF-LiFSI ^g	3.0×10^{-4}	0.44	S10
PVDF-LPPO ^h	4.84×10^{-4}	0.47	S11

PVDF/LLZTO(10%)-CPE ⁱ	5.0×10^{-4}	/	13
PVBL ^j	8.2×10^{-4}	0.57	15
LPE@Ni-DMF	6.5×10^{-4}	0.71	This work

^a PVDF-LiFSI is the abbreviation of polymer-based electrolyte comprised of poly(vinylidene difluoride) (PVDF) and LiFSI. ^b PMLSE is the abbreviation of PVDF-HFP/MOF composite gel/LLZN nanowires solid electrolyte. ^c HSE-etched bmLLZO30 is the abbreviation of hybrid solid electrolytes comprised of 70 wt % PVDF and 30 wt % ball-milled and CF₄ plasma etched LLZO filler. ^d LATP-PVDF-Li is the abbreviation of composite solid electrolyte comprised of LATP powders, PVDF, and LiTFSI. ^e PHLC (20% CAP) is the abbreviation of polymer electrolyte comprised of PVDF-HFP, LiTFSI, and cellulose acetate propionate with a weight ratio of 20% with respect to PVDF-HFP. ^f S-LHCE is the abbreviation of solidified localized high-concentration electrolyte. ^g PVDF-DMF-LiFSI is the abbreviation of solid polymer electrolyte comprised of PVDF, trace DMF solvent, and LiFSI. ^h PVDF-LPPO is the abbreviation of lithium phenyl phosphate grafted onto PVDF. ⁱ PVDF/LLZTO(10%)-CPE is the abbreviation of the PVDF-based CPE membrane with 10 wt % LLZTO fillers. ^j PVBL is the abbreviation of composite solid-state electrolytes constructed by PVDF matrix and BaTiO₃-Li_{0.33}La_{0.56}TiO_{3-x}.

[c. Minor comments]

(c-1) Lines 70–73: To make sure of the authors' claim, it would be nice if the authors could specify the concentration (or weight ratio) of the diluted DMF molecules in Ref. 10 vs. that of another study without using any additive.

Response:

Many thanks for the reviewer's suggestion. Actually, the original Ref. 10 was misquoted, which has been deleted in the revised manuscript, and we apologize for the oversight. We believe that the reviewer expects us to specify the concentration variation of DMF molecules when PAA additive is introduced in Ref. 11 and explain the relationship between diluted DMF solvent and ionic conductivity. As the authors of Ref. 11 showed in their work, the content of residual DMF in the polymer electrolyte with 3 wt % PAA additives was measured to be 21.7 wt %, which is lower than that of the polymer electrolyte without any additive (22.5 wt %). This convincingly demonstrates the diluted DMF molecules when extra component is incorporated into in the polymer electrolyte. Based on this fact, the polymer electrolyte with 5 wt % PAA additives may have the lowest residual DMF content among all the contrast samples in Ref. 11, and indeed it had a lower ionic conductivity (Figure 2a in Ref. 11) than that of the electrolyte with 3 wt % PAA (9.1×10^{-5} S cm⁻¹ at 30 °C). Accordingly, the description has been rephrased in the revised manuscript (**Page 4**) as described below:

“However, despite the improved interfacial stability and cycle life, DMF is virtually diluted and cannot sufficiently dissociate the Li salts when excessive organic additive is added, resulting in compromised ionic conductivity.”

(c-2) Lines 111 and 352: The word “overpotential” would need to be revised as “overvoltage”

because this is a two-electrode measurement and the voltage reading is the difference between the electrode potential between the two Li-metal electrodes.

Response:

Thanks for the reviewer's suggestion. The word “overpotential” has been revised as “overvoltage” in the revised manuscript (**Page 6 and Page 20**) as highlighted below:

“As a result, the designed CPE cycled stably against Li metal electrodes for over 6,000 hours with an **overvoltage** of 64 mV” and “As a result, the Li/LPE@Ni-DMF/Li cell stably cycles for 6,250 hours with a low **overvoltage** of 64 mV, whereas the cell using LPE displays increasing **overvoltage** and interfacial resistance after only 500 hours of cycling.”

(c-3) Lines 119–120: It would be helpful to specify the areal capacity in mAh/cm² as well.

Response:

Many thanks for the reviewer's suggestion. The areal capacity data has been supplemented in the revised manuscript (**Page 6**) as highlighted below:

“These features enable sulfurized polyacrylonitrile//Li (SPAN//Li) cells to operate for 1,000 cycles at 1 C with a capacity decay of 0.04% per cycle and a SPAN//Li pouch cell **with an areal capacity of 1.9 mAh cm⁻²** (47 mAh with sulfur loading of 50 mg) to operate for 35 cycles, all of which stand among the state-of-the-art performance in solid-state lithium–sulfur batteries (ssLSBs).”

(c-4) Line 143: The article “The” would be changed to “This” for better readability.

Response:

We acknowledge the reviewer's careful comment. We have replaced the word of “The” with “**This**” for better readability. (**Page 7**)

(c-5) Line 157: The phrase “Stack date” would be “Stack data”.

Response:

We express our gratitude to the reviewer’s careful examination of our work, and we apologize for the oversight. We have revised the misspelling to “**Stack data**” in the revised manuscript. (**Page 8**)

(c-6) Line 200: In the caption for Fig. 1e, “L-edge” would be “L₃-edge”.

Response:

We acknowledge the reviewer's careful comment. The “L-edge” in the caption for **Fig. 1e** has been revised to “**L₃-edge**”.

(c-7) Line 257: The word “aggregate” would be changed to “concentrate” because “aggregate” would make readers imagine the aggregation (to form big particles).

Response:

Many thanks for the reviewer's suggestion. The word “aggregate” has been revised to “**concentrate**” in the revised manuscript. (**Page 13**)

(c-8) Line 345: The phrase “active anode materials” would be changed to “the anode active material”.

Response:

Many thanks for the reviewer's suggestion. We have rephrased “active anode materials” to “**the**

anode active material” in the revised manuscript. (Page 20)

(c-9) Line 455: I think “up to” is redundant.

Response:

We acknowledge the reviewer's comment. We do agree with the reviewer and we have removed the redundant phrase “up to” in the revised manuscript to improve readability.

(c-10) Line 474: I think it is better to rephrase “LPE@Ni-DMF-SEI” to a clearer explanation (e.g., simply, “the SEI layer” because readers can understand this is the case for LPE@Ni-DMF by reading the clause before “that”).

Response:

We acknowledge the reviewer's careful comment. The phrase “LPE@Ni-DMF-SEI” has been changed to “the SEI layer” in the revised manuscript for better readability. (Page 27)

(c-11) Lines 488–492: It would be nice if the authors could also specify the capacity retention of the SPAN/LPE/Li cell in percentage.

Response:

Many thanks for the reviewer's suggestion. The capacity retention of the SPAN/LPE/Li cell has been supplemented in the revised manuscript (Page 27) as highlighted below:

“...while the SPAN/LPE/Li cell only provides an average specific capacity of 86.5 mAh g⁻¹ at -10 °C, corresponding to a low capacity retention of 13% (Fig. 5e).”

(c-12) Line 541: Maybe the authors would want to add “SPEs for” between “...extended to” and “other emerging solid-state battery systems” because I believe the authors’ approach is for the development of SPEs and is not directly for the emerging solid-state batteries (e.g., electrodes, cell-enclosure, and operational condition designs, etc., which are outside the scope of this study).

Response:

Many thanks for the reviewer's suggestion. We do agree with the reviewer and apologize for the unclear expression. We have revised the corresponding sentence in the revised manuscript (Page 30) as highlighted below:

“...and we expect that the design principle can be extended to SPEs for other emerging solid-state battery systems.”

(c-13) Line 584: Please specify the direction of the experiment (e.g., tested from the highest frequency to the lowest frequency).

Response:

Many thanks for the reviewer's comment. We have rephrased the sentence in the revised manuscript (Page 32) as outlined below:

“...using a Biologic multi-channel electrochemical workstation in the range from high frequency of 100 K Hz to low frequency of 0.1 Hz and an amplitude voltage of 5 mV.”

(c-14) Line 586: The symbol “Ea” should be “E_a” (“a” should be subscripted). Please check the other parts as well.

Response:

We express our gratitude to the reviewer's careful examination of our work, and we apologize for the oversight. The symbol "E_a" has been changed to "E_a". We have checked and revised the manuscript carefully about the issue of subscript.

(c-15) Line 589: The phrase "Li/SSEs/Li cells" would need to be "Li/SSE/Li cells". Please check the other parts as well.

Response:

Many thanks for the reviewer's comment. The phrase "Li/SSEs/Li cells" has been changed to "Li/SSE/Li cells". We also checked the manuscript and made necessary revise that highlighted in the revised manuscript.

(c-16) Line 590: Please cite the reference that proposed this equation.

Response:

Many thanks for the reviewer's suggestion. We have cited one appropriate reference to support our statement as highlighted below:

"Symmetric Li/SSE/Li cells were used to test t_{Li^+} by combining AC impedance and potentiostatic polarization procedures according to the equation⁵: ..." (Page 32)

(c-17) Line 593: The symbol "R_s" should be "R_s" ("s" should be subscripted).

Response:

We acknowledge the reviewer's careful comment. The symbol "R_s" has been changed to "R_s". We have checked and revised the manuscript carefully about the issue of subscript.

(c-18) Lines 609-610: I think the authors would need to specify the test temperatures as they studied multiple temperatures.

Response:

We acknowledge the reviewer's careful comment. The test temperatures have been supplemented in the revised manuscript (Page 33) as highlighted below:

"A LAND CT2003A electrochemical testing system was used to measure the electrochemical performance of symmetrical and asymmetric coin cells at multiple temperatures." and "The cycling performance of the Li/SSE/Li cells was tested at current densities of 0.1 ~ 1 mA cm⁻² at room temperature. The performance of SPAN/SSE/Li cells was tested with the voltage range from 1.0 to 3.0 V at room temperature, 20 °C, 10 °C, 0 °C, and -10 °C, respectively."

(c-19) Lines 624-625: The authors would want to add "(the) thermal stability" here because this is the data that TG analysis can provide but it has not been specified.

Response:

We acknowledge the reviewer's careful comment and apologize for the missing information. The TG analysis has been specified in the revised manuscript ("Materials characterization" section, Page 34) as described below:

"TG (TA instruments Q500) analysis was conducted to probe the thermal stability of SSEs and determine the weight variation of DMF before and after electrochemical tests."

(c-20) Line 697: The symbol "Φ_e" would need to be "Φ_l".

Response:

We express our gratitude to the reviewer's careful examination of our work, and we apologize for the oversight. The symbol " Φ_e " has been revised to " Φ_I ". (Page 37)

REVIEWERS' COMMENTS

Reviewer #1 (Remarks to the Author):

All concerns were fully addressed. Therefore, this work is available for publication.

Reviewer #3 (Remarks to the Author):

The authors have properly revised the manuscript as per the reviewers' suggestions. I think now the manuscript can be recommended for publication. I have been impressed by their convincing point-by-point revision and huge effort to strengthen the significance of the manuscript. I look forward to seeing its published version later.

Before publication, the authors have been requested to check the following minor points to improve the manuscript further:

[Minor Comments]

(1) Supplementary Fig. 35a: What are the white substances with random shapes? Are they impurities, Li dendrites, and/or crystallized Li salts?

(2) Supplementary Fig. 35a: Please add a scale bar.

(3) Line 399: Because of (1), I think we would not be able to say this is "a smooth surface". I suggest deleting "smooth".

(4) The caption of Supplementary Fig. 40: Just in case, the authors would want to explain that the element mappings were conducted in the area surrounded by the white-dotted rectangle (square?) in Supplementary Fig. 40a.

(5) Figs. R7b–R7d: This is out of curiosity, but why do the current values in these figures remain minus? Did they start from 0 (to 5 V) regardless of their OCV values? The LSV should be started from an OCV (or a sufficiently long rest time is needed at the starting voltage), otherwise, the resulting voltammogram will be affected by the electrochemical reactions induced by the voltage jump from the OCV to the starting voltage.

Review Comments for NCOMMS-23-50128A (Round 2)

Point-by-Point Response to the Reviewer's Comments

Reviewer #3 (Remarks to the Author):

The authors have properly revised the manuscript as per the reviewers' suggestions. I think now the manuscript can be recommended for publication. I have been impressed by their convincing point-by-point revision and huge effort to strengthen the significance of the manuscript. I look forward to seeing its published version later.

Response:

We appreciate the reviewer's positive evaluation and support for our work being published in Nature Communications.

Before publication, the authors have been requested to check the following minor points to improve the manuscript further:

[Minor Comments]

(1) Supplementary Fig. 35a: What are the white substances with random shapes? Are they impurities, Li dendrites, and/or crystallized Li salts?

Response:

Actually, we prefer to believe that the white substances with random shapes are accidentally introduced impurities when we disassembled the cycled cell.

(2) Supplementary Fig. 35a: Please add a scale bar.

Response:

Many thanks for the reviewer's reminder. We have added the scale bar in Supplementary Fig. 35.

Supplementary Fig. 35 | (a) SEM image of the Li metal surface after Li/LPE@Ni-DMF/Li symmetric cell testing at room temperature (20 hours of cycling at 0.1 mA cm^{-2} with capacity of 0.1 mAh cm^{-2} per half cycle) and (b) corresponding EDS mappings.

(3) Line 399: Because of (1), I think we would not be able to say this is “a smooth surface”. I suggest deleting “smooth”.

Response:

Many thanks for the reviewer's suggestion. The word “smooth” has been replaced with “flat”, which is more appropriate to describe the cycled Li metal surface.

(4) The caption of Supplementary Fig. 40: Just in case, the authors would want to explain that the element mappings were conducted in the area surrounded by the white-dotted rectangle (square?) in Supplementary Fig. 40a.

Response:

We acknowledge the reviewer's careful review. We have added an explicit description in the caption of Supplementary Fig. 40 as described below:

“Supplementary Fig. 40 | Cryogenic transmission electron microscopy image (a) and corresponding element mappings (b) of LPE@Ni-DMF-SEI. The element mappings were conducted in the area surrounded by the white-dotted square in Supplementary Fig. 40a.”

(5) Figs. R7b–R7d: This is out of curiosity, but why do the current values in these figures remain minus? Did they start from 0 (to 5 V) regardless of their OCV values? The LSV should be started from an OCV (or a sufficiently long rest time is needed at the starting voltage), otherwise, the resulting voltammogram will be affected by the electrochemical reactions induced by the voltage jump from the OCV to the starting voltage.

Response:

Many thanks for the reviewer's suggestion. We have re-tested the LSV of the samples from their OCV values, and there is no obvious redox potential detectable for any of the samples.

Fig. R7 | LSV data of Ni-H₂O tablet (a), Ni-activated tablet (b), and Ni-DMF tablet (c) with a scanning rate of 1 mV s⁻¹.